DISCOVERY REPORT

# An evolutionary conserved detoxification system for membrane lipid–derived peroxyl radicals in Gram-negative bacteria

**Marwa Naguib[1,2]☉, Nicolás Feldman[1]☉, Paulina Zarodkiewicz[1], Holly Shropshire[3], Christina Biamis[1], Omar M. El-Halfawy[4,5], Julia McCain[6], Clément Dezanet[7], Jean-Luc Décout[7], Yin Chen[3], Gonzalo Cosa[6], Miguel A. Valvano ◉[1] ***

**1** Wellcome-Wolfson Institute for Experimental Medicine, Queen's University Belfast, Belfast, United Kingdom, **2** Department of Microbiology and Immunology, Faculty of Pharmacy, Damanhour University, Damanhour, Egypt, **3** School of Life Sciences, University of Warwick, Coventry, United Kingdom, **4** Department of Chemistry and Biochemistry, Faculty of Science, University of Regina, Regina, Saskatchewan, Canada, **5** Department of Microbiology and Immunology, Faculty of Pharmacy, Alexandria University, Alexandria, Egypt, **6** Department of Chemistry and Quebec Center for Advanced Materials, McGill University, Montreal, Québec, Canada, **7** Department of Molecular Pharmacochemistry, Université Grenoble Alpes/CNRS, Grenoble, France

☉ These authors contributed equally to this work.
* m.valvano@qub.ac.uk

The Editors encourage authors to publish research updates to this article type. Please follow the link in the citation below to view any related articles.

## Abstract

How double-membraned Gram-negative bacteria overcome lipid peroxidation is virtually unknown. Bactericidal antibiotics and superoxide ion stress stimulate the transcription of the *Burkholderia cenocepacia bcnA* gene that encodes a secreted lipocalin. *bcnA* gene orthologs are conserved in bacteria and generally linked to a conserved upstream gene encoding a cytochrome $b_{561}$ membrane protein (herein named *lcoA*, lipocalin-associated cytochrome oxidase gene). Mutants in *bcnA*, *lcoA*, and in a gene encoding a conserved cytoplasmic aldehyde reductase (peroxidative stress-associated aldehyde reductase gene, *psrA*) display enhanced membrane lipid peroxidation. Compared to wild type, the levels of the peroxidation biomarker malondialdehyde (MDA) increase in the mutants upon exposure to sublethal concentrations of the bactericidal antibiotics polymyxin B and norfloxacin. Microscopy with lipid peroxidation–sensitive fluorescent probes shows that lipid peroxyl radicals accumulate at the bacterial cell poles and septum and peroxidation is associated with a redistribution of anionic phospholipids and reduced antimicrobial resistance in the mutants. We conclude that BcnA, LcoA, and PsrA are components of an evolutionary conserved, hitherto unrecognized peroxidation detoxification system that protects the bacterial cell envelope from lipid peroxyl radicals.

## Introduction

Aerobic bacteria use molecular oxygen for respiration and oxidation of nutrients with the concomitant production of toxic reactive by-products, which include reactive oxygen species

**Data Availability Statement:** All relevant data are within the paper and its Supporting Information files.

**Funding:** This work was supported by a grant from the Biotechnology and Biological Sciences Research Council (BBSRC grant BB/S006281/1) to MAV, and by grants from the Natural Sciences and Engineering Research Council (NSERC RGPIN-2019-05935) and the Canada Foundation for Innovation (Project Numbers 10401 and 36368) to G.C. MN was supported from a Scholarship Award by the Newton-Moshrafa PhD Programme, a partnership of the British Council and Egypt; N.F. was supported by an International PhD Fellowship from the Northern Ireland Department of Economy; J.M. was supported by an NSERC Post-Graduate Scholarship (CGSD3-519527-2018). OME is supported by a Canada Research Chair in Chemogenomics and Antimicrobial Research (Project Number 950-232965). The funders had no role in study design, data collection and analysis, decision to publish, or preparation of the manuscript.

**Competing interests:** The authors have declared that no competing interests exist.

**Abbreviations:** APG, alanyl-phosphatidylglycerol; CFU, colony-forming unit; CL, cardiolipin; diNn, 3′,6-dinonyl neamine tetratrifluoroacetate; DPPP, diphenyl-1-pyrenylphosphine; FIJI, Fiji Is Just ImageJ; H₄BPMHC, 8-((6-hydroxy-2,5,7,8-tetramethylchroman-2-yl)-methyl)-1,5-di(3-chloropropyl)-pyrromethene fluoroborate; HRP, horseradish peroxide; LB, Luria-Bertani; LPS, lipopolysaccharide; MDA, malondialdehyde; MIC, minimum inhibitory concentration; NAO, Acridine Orange 10-nonyl bromide; NBD-Pen, 2,2,6-trimethyl-4-(4-nitrobenzo [1,2,5]oxadiazol-7-ylamino)-6-pentylpiperidine-1-oxyl; PE, phosphatidylethanolamine; PG, phosphatidylglycerol; ROS, reactive oxygen species; SD, standard deviation; SPE, sphingosyl phosphatidylethanolamine; TBH, *tert*-butyl hydroperoxide.

(ROS), such as superoxide anion radicals, hydrogen peroxide, and hydroxyl radicals [1], as well as nitrogen and electrophilic species. Oxidative stress may also arise from bacterial exposure to physical agents (e.g., ultraviolet radiation), metals, and chemicals (e.g., potassium tellurite, paraquat, and bactericidal antibiotics) [1–3]. Free radicals damage cell membranes via the formation of lipid peroxides. Lipid peroxidation is a nonenzymatic autocatalytic radical chain reaction resulting in the destruction of the phospholipid acyl chains and production of toxic reactive aldehydes such as acrolein, 4-hydroxynonenal, and malondialdehyde (MDA), and eventually alters membrane fluidity and barrier function [4–7]. Like oxygen radicals, lipid aldehydes can react with DNA and proteins, but they are more stable and can cause sustained damage [1].

Gram-positive bacteria contain a single cell membrane surrounded by a thick cell wall peptidoglycan layer, while the Gram-negative bacterial cell envelope consists of 2 membranes, inner and outer membrane, separated by a periplasmic region containing cell wall peptidoglycan. The inner membrane is a phospholipid bilayer; the outer membrane is an asymmetrical bilayer consisting of phospholipids at the inner leaflet and lipopolysaccharide (LPS) at the outer leaflet [8]. The most abundant phospholipids in Gram-negative bacteria are phosphatidylethanolamine (PE), phosphatidylglycerol (PG), and cardiolipin (CL), all carrying monounsaturated acyl chains [9–11]; the lipid A moiety of the LPS generally contains saturated acyl chains [8]. Upon oxidative stress, bacteria produce antioxidant enzymes (e.g., superoxide dismutases and catalases/peroxidases) that attenuate ROS-generated oxidative damage [12,13]. The response to ROS has been well studied [14], but how bacterial cells, especially the double-membraned Gram-negatives, overcome membrane lipid peroxidation stress is unclear. This partly relates to the prevailing notion that membrane lipids containing polyunsaturated fatty acids with labile bis-allylic hydrogen atoms, which are highly susceptible to lipid peroxidation [15], are generally absent from bacteria, whose membranes are rich in poorly oxidizable saturated or monounsaturated lipid molecules [10,16,17]. However, lipid peroxidation occurs in bacteria under many conditions, such as exposure to *tert*-butyl hydroperoxide (TBH) [18], hydrogen peroxide [19], potassium tellurite [2,20] and titanium oxide [21], defects in the synthesis of polyamines [22], and sublethal concentrations of antibiotics [3,23,24]. Because of the destructive nature of the lipid peroxidation reaction, bacterial cells must deal with both lipid radical formation and membrane repair. While restoring oxidized lipids in the inner membrane (or the cell membrane in Gram-positives) would be expected through regulatory responses influencing phospholipid synthesis, a mechanism to protect Gram-negative bacterial cells against lipid peroxyl radicals and toxic reactive aldehydes arising from the peroxidation of outer membrane phospholipids is presently unknown.

In addition to conventional mechanisms of antibiotic resistance, we previously discovered that the opportunistic human pathogen *Burkholderia cenocepacia* can resist bactericidal antibiotics by mechanisms operating extracellularly [25–27]. Key molecules involved in these mechanisms are the polyamine putrescine [26] and the BcnA lipocalin protein [27]. BcnA, which belongs to the widely conserved YceI family of bacterial lipocalins, protects bacteria from bactericidal antibiotics by molecular scavenging [27]. *B. cenocepacia* also produces BcnB, another lipocalin that lacks antibiotic binding activity [27]. BcnA and BcnB proteins share a characteristic lipocalin β-barrel shape [27,28] composed of 8 antiparallel β-strands and relatively unstructured (flexible) loops at the open end, which form a "cup" domain [27]. In solution, BcnB forms a dimer, while BcnA is a monomeric protein. Both proteins are predicted to be in the periplasm, but only BcnA is secreted extracellularly in *B. cenocepacia* by an unknown mechanism [27]. Molecular dynamics and in silico docking experiments revealed that antibiotics interact with the rim of BcnA's cup domain [27]; their binding interactions are weaker compared to that of hydrophobic ligands that can access the interior of the tunnel such as Nile

red, polymyxin B, and α-tocopherol [27,29]. The structural and molecular binding characteristics of BcnA argue against the notion that antibiotic binding is the primary function of this lipocalin.

In addition to BcnA and BcnB (PDB IDs 5IXH and 5IXG, respectively) [27], several YceI bacterial lipocalins have been crystallized from *Escherichia coli* (PDB ID 1Y0G), *Pseudomonas syringae* (PDB ID 3Q34), *Campylobacter jejuni* (PDB ID 2FGS), *Helicobacter pylori* (PDB ID 3HPE) [30], *Pseudomonas aeruginosa* (PDB ID 7BWL) [31], *Thermus thermophilus* (PDB ID 1WUB) [32], *Saccharophagus degradans* (PDB ID 2X32) [33], and *Treponema pallidum* (PDB ID 5JK2) [34]. These molecules reveal remarkable structural conservation despite notable differences in their amino acid sequences. Further, an isoprenoid molecule, typically octaprenyl-phosphate, octaprenylphenol, or ubiquinone-8 (Ubi-8, coenzyme Q), has been detected in the lipocalin β-barrel tunnel of most of the structures. From these, Ubi-8 plays a role in electron transfer reactions and as an antioxidant [35,36]. The observation that the *Neisseria meningitidis* YceI lipocalin has Ubi-8 bound when purified from periplasmic extracts [37] argues that Ubi-8 could be a natural substrate for YceI lipocalins.

The *B. cenocepacia bcnA* and *bcnB* genes are in a 3-gene operon; they are located downstream from a gene encoding a predicted membrane cytochrome $b_{561}$ protein [27], which we herein named lipocalin-associated cytochrome oxidase gene (*lcoA*; S1 Fig). We previously showed that *lcoA* (formerly *bcoA*), *bcnA*, and *bcnB* are cotranscribed [29]; *lcoA* and *bcnA* are also up-regulated in response to sublethal concentrations of bactericidal antibiotics and the pro-oxidant herbicide paraquat [25,27]. Similarly, the transcription of the *P. aeruginosa* PA0423 *bcnA* ortholog also increases in response to hydrogen peroxide and paraquat [38]. The physical and regulatory link between *bcnA* and *lcoA* suggests that their protein products could provide an antioxidant function. Due to the periplasmic and membrane locations of BcnA and LcoA, respectively, we hypothesized that these proteins are involved in protecting outer membrane phospholipids from peroxyl radicals, particularly under oxidative stress mediated by sublethal concentrations of antibiotics and other pro-oxidant molecules, such as paraquat or toxic metals. In this study, we demonstrate that both BcnA and LcoA, together with the conserved cytoplasmic aldehyde reductase PsrA, are components of a hitherto unrecognized system to protect Gram-negative bacteria from membrane lipid–derived peroxidation damage.

## Results

### *bcnA* and *lcoA* gene orthologs are genetically linked in many different Gram-negative bacterial species

To support the notion that BcnA and LcoA proteins could be functionally linked, we examined the synteny of their respective gene orthologs in the β-proteobacteria using MultiGeneBlast [39]. The results revealed that *lcoA-bcnA-bcnB* gene orthologs are present in the same organization in most species of the β-proteobacteria, including all members of the *Burkholderia* genus (S1 Fig). Exceptions were found in *Neisseria*, *Bordetella*, and *Achromobacter* species, where *bcnA* was identified as a monocistronic gene unlinked to *lcoA* or *bcnB* (S1 Fig). The *lcoA-bcnA* genes are also conserved as a predicted 2-gene operon in some of the γ-proteobacteria (e.g., *Escherichia*, *Enterobacter*, *Proteus*, and *Pseudomonas*, while *bcnA* gene orthologs are monocistronic in others (e.g., *Klebsiella* and *Acinetobacter*) (S1 Fig). A conserved gene organization within bacterial operons is generally related to function [40]. In most of the α-proteobacteria (e.g., *Caulobacter*, *Agrobacterium*, *Sphingomonas*, *Rhizobium*, and *Parvularcula*), *lcoA* and *bcnA* genes are fused, suggesting that they encode a predicted chimeric protein with a cytochrome $b_{561}$ amino-terminal domain and a BcnA lipocalin carboxyl-terminal domain. In these cases, the predicted LcoA-BcnA chimeras have a linker domain that replaces the amino

acids corresponding to the signal peptide of the lipocalin, suggesting that BcnA is tethered to the membrane-embedded cytochrome $b_{561}$. The conserved gene neighborhood and examples of *lcoA-bcnA* fused genes imply a functional link between the LcoA cytochrome $b_{561}$ and the BcnA lipocalin.

## The absence of BcnA, LcoA, and PsrA is associated with increased levels of membrane lipid peroxidation

To test whether BcnA and LcoA participate in membrane lipid–mediated peroxidation stress, we evaluated the membrane lipid peroxidation status of the wild-type bacterium *B. cenocepacia* K56-2 and its isogenic *bcnA* and *lcoA* deletion mutants using diphenyl-1-pyrenylphosphine (DPPP). DPPP is a nonfluorescent molecule that partitions into lipid bilayers and reacts stoichiometrically with hydroperoxides in the membrane giving a fluorescent DPPP oxide (DPPP = O) [41,42]. The fluorescence intensity of DPPP = O was measured in bacteria challenged with minimum inhibitory concentration (MIC) levels of potassium tellurite for 1 hour to induce oxidative stress [2,20] (Fig 1A). The fluorescence intensities of DPPP = O recorded from untreated Δ*bcnA* and Δ*bcnAB*Δ*lcoA* mutants were 2.5-fold higher than in K56-2 and the other mutants, suggesting that loss of *bcnA* associates with a higher steady state of membrane lipid peroxidation. In contrast, upon tellurite challenge, the fluorescence intensity in all the mutants lacking either *bcnA* or *lcoA* was significantly higher than in K56-2, while the DPPP = O fluorescence maintained wild-type levels when plasmids encoding either BcnA or LcoA were introduced into the corresponding mutants. No significant differences relative to K56-2 were found in the peroxidation levels of the Δ*bcnB* mutant ($p = 0.6$).

Studies in *E. coli* have identified *yqhD*, a gene involved in the response to tellurite-induced oxidative stress. This gene encodes a cytoplasmic aldehyde reductase that detoxifies reactive aldehydes resulting from membrane lipid peroxidation by catalyzing the NADPH-dependent reduction of various membrane peroxidation-derived short-chain aldehydes, including acrolein and MDA [2]. *yqhD* orthologs are conserved in thousands of bacterial genomes but unlinked to *bcnA* and *lcoA*. Strain K56-2 contains an *yqhD* ortholog, *K562_20604* (GenBank accession CP053301), which we renamed peroxidative stress-associated aldehyde reductase gene (*psrA*). To evaluate whether PsrA contributes to reduce peroxidation levels, we constructed a Δ*psrA* mutant, which was examined for peroxidation with DPPP under tellurite challenge. In unchallenged Δ*psrA* bacteria, the peroxidation levels were comparable to those of K56-2, but the peroxidation levels in the mutant increased significantly under tellurite stress (Fig 1A, S1 Data). These results suggest the accumulation of short aldehydes inducing additional membrane lipid peroxidative stress. Further examination of the PsrA polypeptide by de novo homology modeling revealed a predicted structure that overlaps to that of the *E. coli* YqhD protein (PDB ID 1OJ7) despite both proteins only sharing 21% amino acid identity (Fig 2A). Importantly, a perfect conservation of the catalytic sites Asp-193, His-197, His-266, and His-280 (numbering from YqhD) was noted (Fig 2A). Analysis of membrane lipid peroxidation levels using DPPP showed a significant decrease in the Δ*psrA* mutant when complemented via a plasmid encoding the aldehyde reductase of either *E. coli* or *B. cenocepacia* K56-2 (Fig 2B, S2 Data). Moreover, challenge experiments detecting the peroxidation by-product MDA upon treatment with norfloxacin and polymyxin B demonstrated that the Δ*psrA* mutant can be functionally complemented with the *E. coli* YqhD at similar levels as with the *B. cenocepacia* PsrA putative aldehyde reductase (Fig 2C, S2 Data). From these results, we conclude that PsrA and YqhD are functionally identical orthologs.

Next, we investigated whether mutants unable to produce BcnA, LcoA, and PsrA can overcome physical stress conditions known to cause peroxidation. One such condition is cold

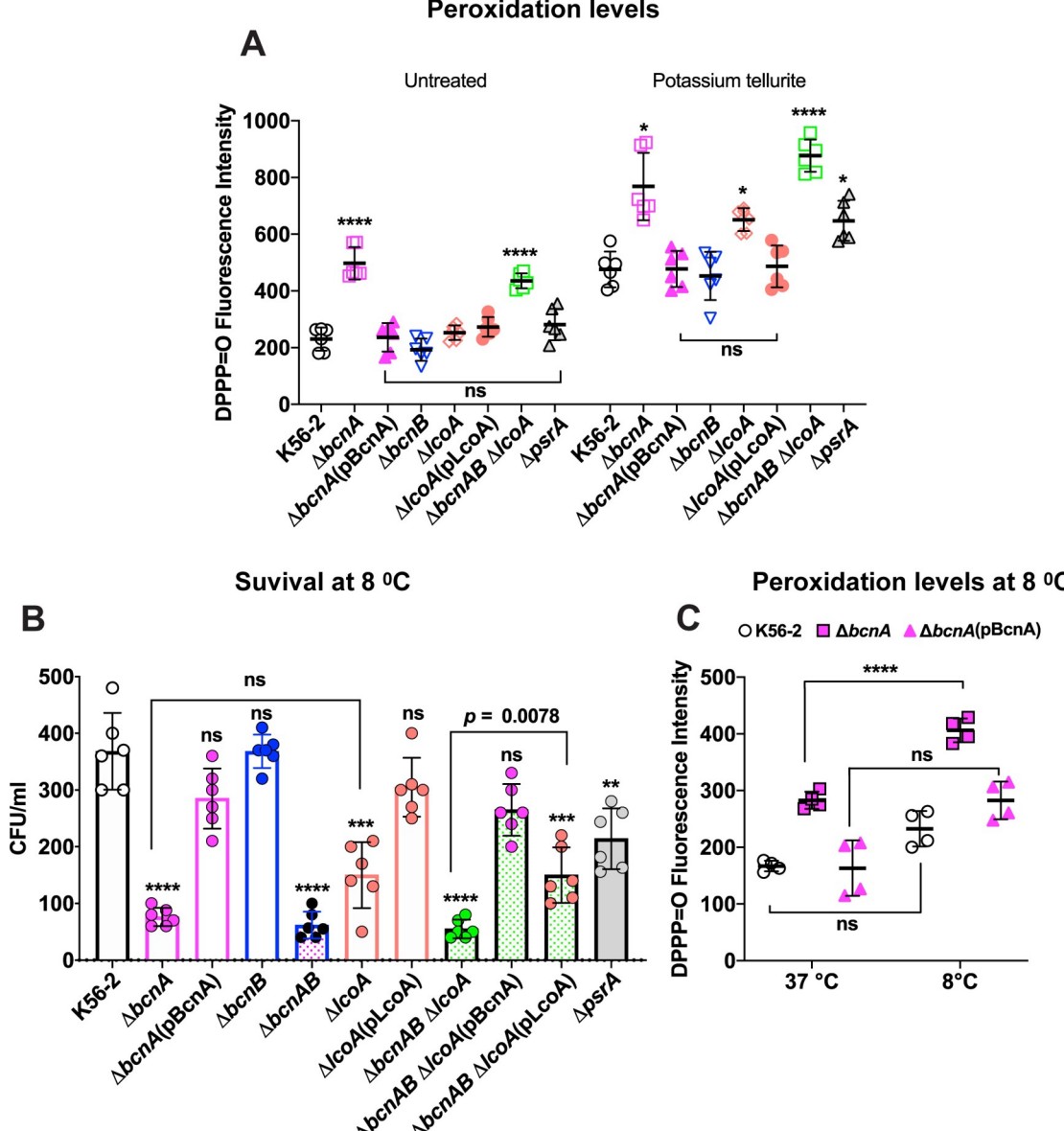

**Fig 1. BcnA, LcoA, and PsrA help maintain reduced levels of membrane lipid peroxidation under metal- and low temperature–induced stress.** K56-2 and mutants carried the control plasmid vector pDA17 or the indicated complementing plasmid (in parentheses). **(A)** Peroxidation levels under potassium tellurite stress. Each bacterial strain was challenged with potassium tellurite at concentrations corresponding to the MIC (50 μg ml$^{-1}$ for K56-2 and ΔbcnB, 25 μg ml$^{-1}$ for ΔpsrA, ΔbcnA (pBcnA), ΔlcoA and ΔlcoA(pLcoA), and 12.5 μg ml$^{-1}$ for ΔbcnA and ΔbcnABΔlcoA) for 1 hour in the presence of DPPP. Lipid peroxidation was assessed fluorometrically by measuring the level of DPPP = O (Materials and methods). Results are shown as the mean of DPPP = O fluorescence ± SD. Data represent the results of 3 independent experiments, each done in duplicate. The statistical significance of fluorescence levels obtained from tellurite-treated versus untreated bacteria was determined by 2-way ANOVA. Strains comparisons were made against K56-2 by Dunnett post hoc analysis. *, $p \leq 0.05$; ****, $p \leq 0.0001$, ns, nonsignificant. **(B)** Bacterial survival at cold temperature. Bacterial cultures were diluted to $10^3$ CFU/ml and left at 8°C for 24 hours at which time the remaining CFU/ml were determined. Results are shown as the mean CFU/ml ± SD of surviving bacteria from 3 independent biological replicates in duplicate. $p$-Values relative to the CFU/ml recovered from K56-2 were calculated by the 1-way Welch ANOVA test with Dunnett post hoc analysis. **, $p \leq 0.01$; ***, $p \leq 0.01$ ****, $p \leq 0.0001$, ns, nonsignificant ($p = 0.28$ for ΔbcnA(pBcnA)), $p > 0.99$ for ΔbcnB, $p = 0.55$ for ΔlcoA(pLcoA), and $p = 0.09$ for ΔbcnABΔlcoA(pBcnA)). Comparison of the recovery of ΔbcnA versus ΔlcoA is not statistically significant ($p = 0.2032$). **(C)** Peroxidation levels measured by DPPP oxidation of K56-2, ΔbcnA, and ΔbcnA(pBcnA) incubated at 8°C as indicated in (B) or at 37°C. The results represent the mean ± SD evaluated by unpaired $t$ test from 2 independent experiments. ****, $p < 0.0001$; ns, nonsignificant. Data underlying the graphs in this figure can be found in S1 Data. CFU, colony-forming unit; DPPP, diphenyl-1-pyrenylphosphine; MIC, minimum inhibitory concentration; *psrA*, peroxidative stress-associated aldehyde reductase gene; SD, standard deviation.

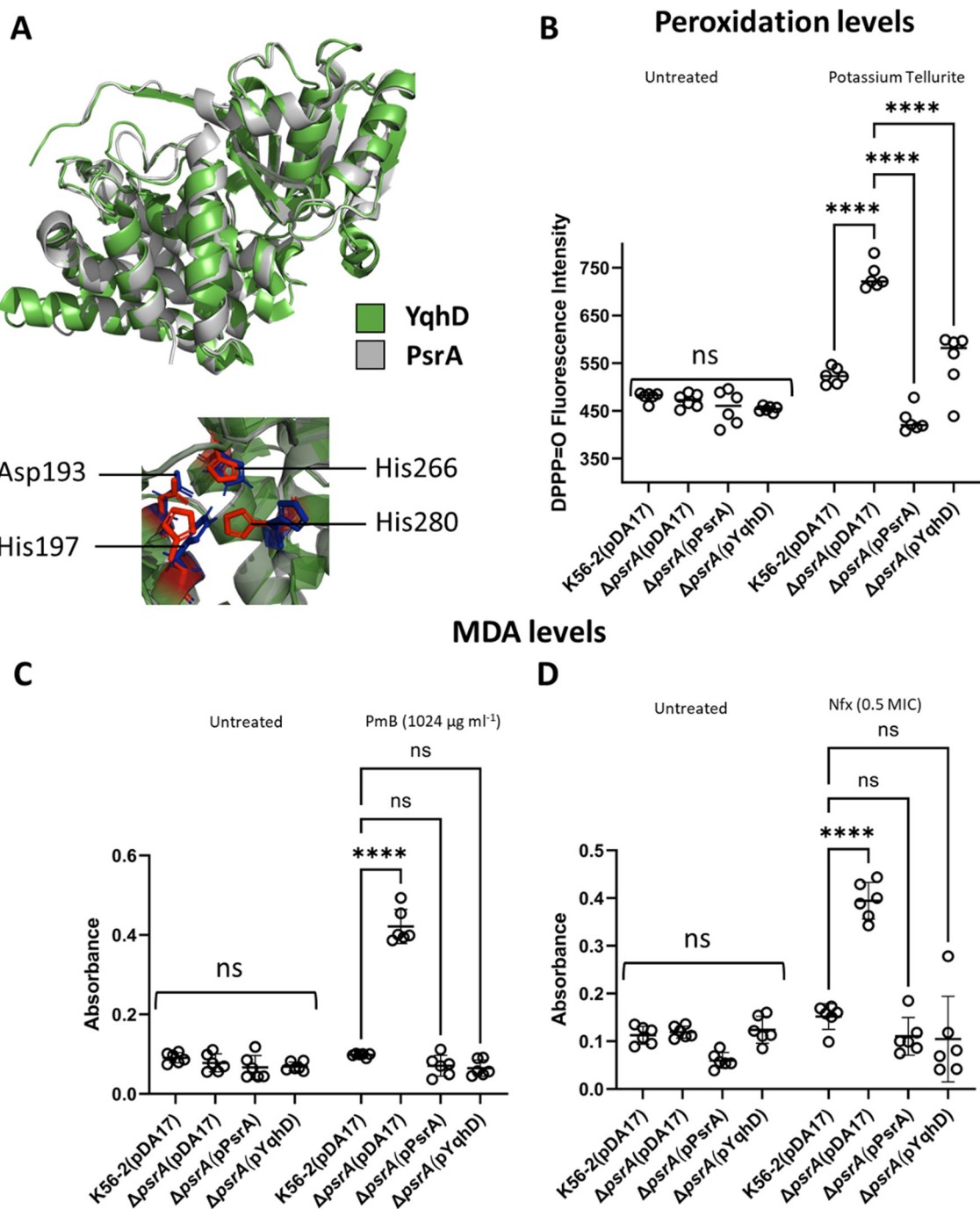

**Fig 2. PsrA and YqhD are orthologs sharing structural and functional properties. (A)** Overlapping of the overall structures and catalytic regions of PsrA and YqhD. The amino acid numbering in the catalytic site correspond to that of YqhD. YqhD and PsrA chains are shown in green and gray, respectively. Catalytic residues if the active site of YqhD and PsrA are indicated in red and blue, respectively. **(B)** Membrane lipid peroxidation levels under metal stress. K56-2 and Δ*psrA* contained either the control plasmid vector pDA17 or the indicated complementing plasmid (in parentheses). Each bacterial strain was challenged with potassium tellurite at concentrations corresponding to the MIC (50 μg ml⁻¹ for the wild-type strain K56-2, Δ*psrA*(pPsrA), Δ*psrA*(pYqhD) and 25 μg ml⁻¹ for Δ*psrA*(pDA17) for 1 hour in the presence of DPPP. **(C and D)** Detection of the lipid peroxidation by-product MDA upon overnight with or without 1,024 μg ml⁻¹ PmB (panel C) or 32 μg ml⁻¹ norfloxacin (Nfx) (panel D). The MDA levels were normalized to bacterial protein concentrations. K56-2 and Δ*psrA* contained the control plasmid vector pDA17 or the indicated complementing plasmid (in parentheses). In panels B through D, results are shown as the mean of absorbance or fluorescence ± SD. Data represent the results of 3 independent experiments, each done in duplicate. The statistical significance of MDA and DPPP = O levels obtained from treated versus untreated bacteria was determined by 2-way ANOVA. Strains comparisons were made against

the wild-type K56-2 by Dunnett multiple comparison test. ****, $p \leq 0.0001$; ns, nonsignificant. Data underlying all graphs in this figure can be found in S2 Data. DPPP, diphenyl-1-pyrenylphosphine; MDA, malondialdehyde; MIC, minimum inhibitory concentration; PmB, polymyxin B; *psrA*, peroxidative stress-associated aldehyde reductase gene; SD, standard deviation.

stress, which increases lipid peroxidation in plants and involves lipocalins in thermotolerance [43,44]. Compared to the wild-type strain, Δ*bcnA*, Δ*lcoA*, and Δ*bcnAB*Δ*lcoA* mutants showed significant reduction in bacterial recovery ($p \leq 0.0001$) when incubated at 8˚C for 24 hours (Fig 1B). The bacterial recovery reached levels like the wild-type strain upon complementation of Δ*bcnA* and Δ*lcoA* mutants with the corresponding plasmids restoring BcnA and LcoA function, respectively. It should be noted that because these mutants are in-frame gene deletions, *lcoA* should be functional in Δ*bcnA*, and, conversely, *bcnA* should be functional in Δ*lcoA*. However, the Δ*bcnAB*Δ*lcoA* recovered similar growth levels as wild type only when complemented with pBcnA, while complementation with pLcoA afforded partial restoration of growth (note that the difference in bacterial recovery between Δ*bcnAB*Δ*lcoA* and Δ*bcnAB*Δ*lcoA*(pLcoA) was statistically significant; $p = 0.0078$, Fig 1B). Together, these results suggest that both BcnA and LcoA are required for protection against cold stress, with BcnA appearing to play a more dominant role. This could be attributed to the presence of additional cytochrome $b_{561}$ paralogs in *B. cenocepacia*. Alternatively, it is also possible that the differences in complementation may be due to different levels of protein expression from recombinant plasmids, a phenomenon we have observed before with complementation experiments in *B. cenocepacia* [45]. The results also showed that Δ*psrA* causes significant reduction in growth recovery after cold stress. As with the previous experiments (Fig 1A), no significant differences were detected in the survival of Δ*bcnB* compared to wild type (Fig 1B). We experimentally validated that cold treatment induces membrane lipid peroxidation by measuring DPPP = O fluorescence in the wild type and the Δ*bcnA* mutant under the same conditions used for the survival experiments (Fig 1C); the results confirmed that the level of lipid peroxidation for Δ*bcnA* under cold stress is significantly higher than the levels in the wild-type strain K56-2 and in the Δ*bcnA* mutant grown at 37˚C. Therefore, reduced bacterial survival in the cold of mutants lacking *bcnA*, *lcoA*, and *psrA* genes singly or in combinations was associated with increased membrane lipid peroxidation.

We also performed oxidative challenge experiments to ascertain whether the loss of BcnA, LcoA, and PsrA proteins influence bacterial susceptibility to oxidative chemical stress. In previous work, we showed that hydrogen peroxide is detoxified by *B. cenocepacia* KatA and KatB catalases [46,47]. Therefore, we used TBH as a catalase-resistant analogue of hydrogen peroxide that generates free radicals in the presence of copper or iron ions [44,48] and promotes lipid peroxidation. Exposing bacteria for 1 hour to 30 μM TBH caused a significant reduction in bacterial survival of Δ*bcnA* relative to K56-2 ($p = 0.0006$) (S2 Fig). Similar results were obtained by challenging the Δ*bcnAB*Δ*lcoA* and Δ*bcnAB*Δ*lcoA*Δ*psrA* mutants ($p = 0.0009$ and $p = 0.0007$, respectively), both of which have in common the absence of *bcnA*. In contrast, Δ*lcoA*, Δ*psrA*, and Δ*bcnB* mutants (all containing a functional BcnA protein), as well as the genetically complemented strains Δ*bcnA*(pBcnA) and Δ*bcnAB*Δ*lcoA*(pBcnA), showed no significant differences in colony-forming units (CFUs) upon TBH challenge compared with the wild-type strain K56-2. We conclude that a functional BcnA is critical for protection against TBH challenge, while complementation of the Δ*bcnAB*Δ*lcoA* mutant with the plasmid encoding the LcoA protein showed an intermediate restoration in CFU counts upon TBH exposure, indicating that the LcoA cytochrome protein can also contribute partial protection if the BcnA protein is absent.

The combined results of all the experiments described above argue that BcnA, LcoA, and PsrA are involved in the adaptation of *B. cenocepacia* to chemical and physical stress cues that lead to membrane lipid peroxidation. In contrast, the *bcnB* gene does not play any role in membrane lipid peroxidation under the investigated conditions.

## BcnA, LcoA, and PsrA show oxygen-dependent increased accumulation of lipid peroxidation by-products upon antibiotic-mediated stress

The involvement of BcnA, LcoA, and PsrA in reducing the level of membrane lipid peroxidation suggested that mutants in these genes accumulate lipid peroxidation by-products. At sublethal doses, bactericidal antibiotics (e.g., β-lactams, quinolones, and colistin/polymyxin B) induce the formation of hydroxyl radicals that oxidize lipid membranes leading to the accumulation of reactive toxic aldehydes such as MDA [3]. The levels of MDA therefore serve as a proxy to assess the magnitude of lipid peroxidation and toxic aldehydes production. We pretreated bacteria with 2 different bactericidal antibiotics, norfloxacin and polymyxin B, at concentrations equivalent to half their respective MICs. The results using norfloxacin (S3A Fig) show that in the absence of the BcnA protein, MDA levels increased by 3-fold compared to wild type under exposure to sublethal concentration of norfloxacin (S3A Fig, $p < 0.0001$). MDA levels returned to wild-type concentrations upon genetic complementation ($p = 0.99$). Similarly, a statistically significant increase in the levels of MDA was seen in Δ*lcoA* mutant versus the wild-type strain ($p = 0.02$), which returned to wild-type levels upon complementation with LcoA. In the case of Δ*psrA*, there was also a significant 2-fold increase in the level of modified protein in comparison to wild type ($p < 0.001$). As with the DPPP experiments, there was no difference in MDA levels between wild type and Δ*bcnB*. Similar results were obtained in bacteria incubated with polymyxin B (S4 Fig). From these experiments, we concluded that bacteria lacking BcnA, LcoA, and PsrA accumulate the lipid peroxidation by-product MDA upon antibiotic-induced stress.

We reasoned that if BcnA, LcoA, and PsrA are required during peroxidation stress, these proteins would be dispensable under reduced concentrations of molecular oxygen. Bacteria were challenged with 0.25 MIC of norfloxacin, and surviving cells were enumerated after growth under microaerophilic conditions that allow survival of *B. cenocepacia*, for over 48 hours [49]. Control experiments showed that *B. cenocepacia*, which is an obligate aerobic bacterium, did not grow under these microaerophilic conditions with concentrations of norfloxacin above 0.25 MIC. No significant differences were found in the survival of wild-type and mutant strains in the absence or in the presence of 0.25 MIC norfloxacin (S3B Fig). Despite that in the presence of antibiotic the number of surviving bacteria was lower than in the untreated control (probably due the added stress of the antibiotic treatment under microaerophilia), no differences were observed among the strains (S3B Fig), indicating that both wild type and mutants exhibit similar antibiotic susceptibility under microaerophilic conditions. In contrast, the control experiment under aerobiosis revealed differences in norfloxacin susceptibility, consistent with the loss of BcnA, LcoA, and PsrA function (S3C Fig). Indeed, all mutants lacking the *bcnA* gene showed significant reduction in the number of surviving bacteria relative to wild type ($p < 0.0001$). Similar results were obtained with the single Δ*lcoA* and Δ*psrA* mutants ($p < 0.0001$ and $p < 0.001$, respectively). Bacterial growth was restored to wild type or near wild-type levels in mutants genetically complemented with either BcnA or LcoA encoding plasmids. As before, the Δ*bcnB* mutant treated with norfloxacin displayed no difference in susceptibility when compared to wild type ($p = 0.79$). From these results, we conclude that the function of BcnA, LcoA, and PsrA proteins are not required under reduced oxygen levels, consistent with their role in reducing the accumulation of lipid peroxidation products under oxidative stress.

## BcnA, LcoA, and PsrA promotes reduced levels of cell envelope peroxidation especially at the cell poles and the septum

We directly assessed the level of membrane peroxidation by fluorescence microscopy using 2,2,6-trimethyl-4-(4-nitrobenzo [1,2,5]oxadiazol-7-ylamino)-6-pentylpiperidine-1-oxyl (NBD-Pen), a fluorogenic probe that becomes emissive [50] upon scavenging lipid peroxyl radicals in living cells due the presence of an α-substituted nitroxide [51,52]. In these experiments, we compared the Δ*bcnA* mutant and the wild-type strain K56-2, both without and with antibiotic stress caused by exposure to sub-MIC of norfloxacin (4 μg ml$^{-1}$ and 48 μg ml$^{-1}$, respectively [29]) over 4 hours before adding the fluorescent probe. Fluorescence images of untreated cells showed that K56-2 bacteria fluoresced with lower intensity than Δ*bcnA* (Fig 3A). In contrast, the Δ*bcnA* cells treated with norfloxacin show higher levels of fluorescence around cells, with a punctate distribution at the poles and in some cells at the midbody, corresponding to cell division sites. K56-2 cells treated with norfloxacin also displayed discrete fluorescence patches at the poles, with some cells lacking detectable peroxidation when comparing fluorescence and phase contrast images. K56-2 bacterial cells treated with norfloxacin gave a fluorescence intensity comparable to that of the Δ*bcnA* control without norfloxacin, further supporting the idea that at steady state, bacteria lacking BcnA produce higher levels of peroxidation. Quantification of fluorescence (Fig 3B) indicated significant differences between norfloxacin-treated and norfloxacin-untreated bacteria, as well as significant differences between norfloxacin-treated K56-2 and Δ*bcnA*, supporting the notion that BcnA has a role in reducing lipid peroxyl radical formation.

We also examined the level of membrane peroxidation by quantitative fluorescence in the Δ*lcoA* and Δ*psrA* mutants. Unlike Δ*bcnA*, Δ*lcoA* produced higher levels of membrane lipid peroxidation in the absence of antibiotic compared to the wild-type K56-2 treated with norfloxacin (S5A Fig). We speculate that the Δ*lcoA* mutant has a higher level of basal lipid peroxidation arising from an increased intracellular level of ROS [53]. The Δ*psrA* mutant, in contrast, behave like Δ*bcnA* by showing peroxidation levels in the untreated condition that were like those of antibiotic treated K56-2. Quantification of fluorescence (S5B Fig) demonstrated significant differences between treated and untreated bacteria, as well as a significant difference between the wild-type strain K56-2 and the mutants, Δ*lcoA*, and Δ*psrA*. Similar results were obtained using polymyxin B for all the mutants (S6 and S7 Figs). To confirm that lipid peroxyl radicals at the cell poles are more abundant under antibiotic stress, we quantified the fluorescence intensity across the long axis of the cells. The results show that, in contrast to cells under no antibiotic stress, where the fluorescence is homogenous along the bacterial cell length, peaks of higher intensity are present at the poles when the bacteria are exposed to antibiotics (S8 Fig).

Given that nitroxides and their fluorogenic analogues are radical trapping antioxidant scavengers both in solution and in micelles [54,55], the emission intensity enhancement with nitroxide–fluorophore adducts may partly arise because of radical trapping antioxidant scavenging in solution, rather than scavenging lipid peroxyl radicals [56]. Therefore, we also investigated membrane lipid peroxidation with the lipophilic fluorogenic antioxidant probe 8-((6-hydroxy-2,5,7,8-tetramethylchroman-2-yl)-methyl)-1,5-di(3-chloropropyl)-pyrromethene fluoroborate (H$_4$BPMHC) [56–58], which consists of α-tocopherol chromanol moiety (PMHC, a chromanol-based free radical scavenger) and a BODIPY lipophilic fluorophore that renders the lipophilic character of the fluorogenic probe and facilitates its incorporation into membranes. The α-tocopherol chromanol moiety in H$_4$BPMHC quenches the BODIPY fluorescence via photo-induced electron transfer until this moiety is oxidized by lipid peroxyl radicals, restoring fluorescence, and allowing the generation of these lipid peroxyl radicals to be

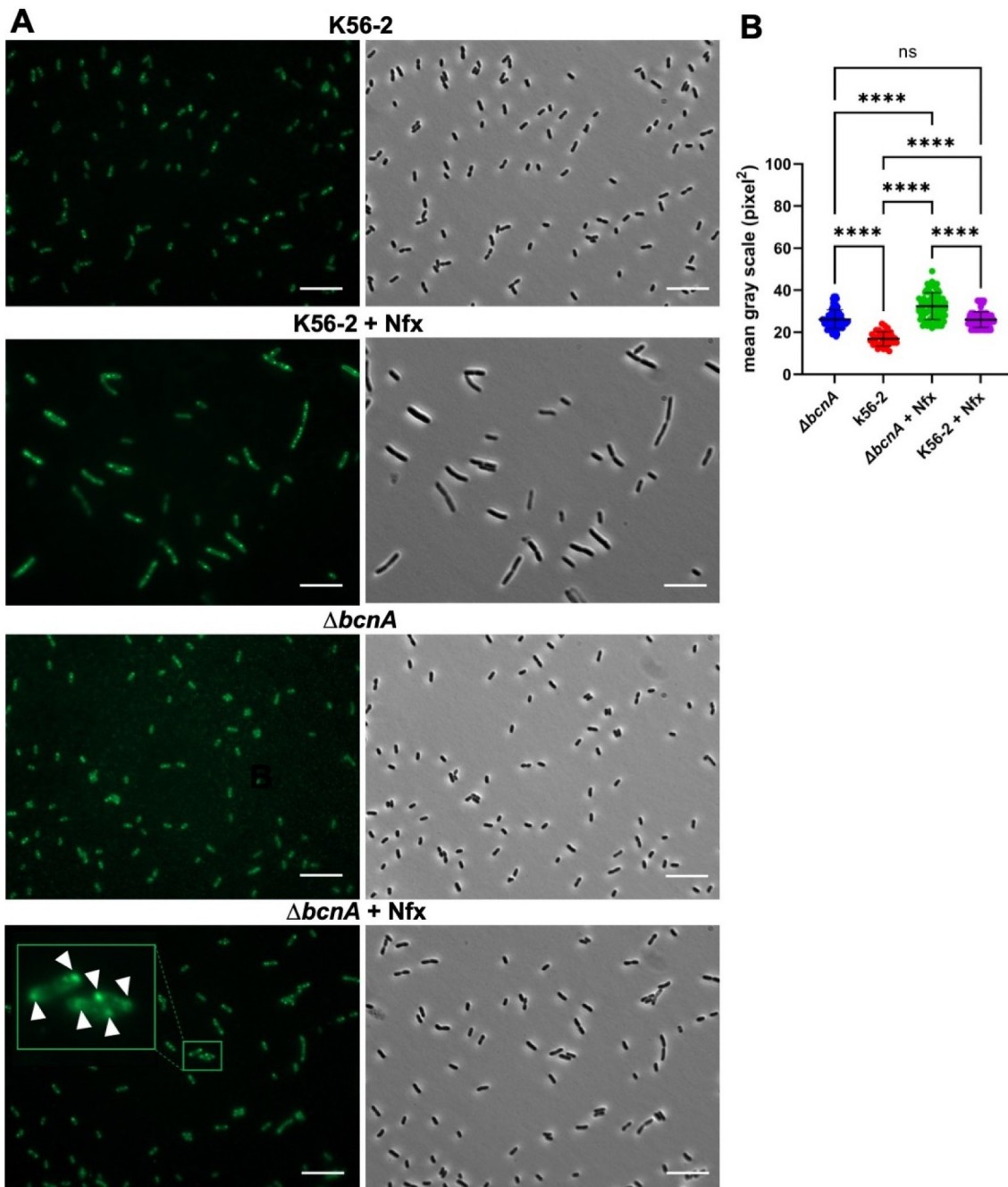

**Fig 3. Detection of lipid peroxidation in Δ*bcnA* by fluorescence microscopy using NBD-Pen. (A)** Fluorescence (left) and phase contrast (right) images of Δ*bcnA* and K56-2 strains untreated or treated with 75% MIC for norfloxacin (Nfx) and incubated with NBD-Pen. The insets are zoomed sections of the images to provide more detail of the location of the fluorescent at the cell poles and mid cell (arrowheads). Scale bars, 10 μm. **(B)** Quantitative analysis of fluorescence intensity based on fluorescence images in panel A. SD shown as error bars, and mean values shown as horizontal black lines. Statistical significance was determined by 1-way ANOVA with Tukey multiple comparisons test post hoc analysis. ****, $p < 0.0001$; ns, nonsignificant. Data underlying the graphs in this figure can be found in S2 Data. MIC, minimum inhibitory concentration; NBD-Pen, 2,2,6-trimethyl-4-(4-nitrobenzo [1,2,5]oxadiazol-7-ylamino)-6-pentylpiperidine-1-oxyl; SD, standard deviation.

visualized [56–58]. In this experiment, the fluorescence was seen in the cell envelope of the mutant *bcnA* at its normal state, correlating with peroxidation of membrane lipids (S9A Fig). The distribution of fluorescence adopted a punctate pattern with strong accumulation of fluorescent spots at bacterial cell poles and septa. Image quantification demonstrated that Δ*bcnA* had 3-fold higher fluorescence intensity than K56-2 ($p = 0.0001$) and the complemented mutant Δ*bcnA*(pBcnA) (S9B Fig). We therefore conclude that the localization of the peroxidation in the membrane was independent of the chemistries of the fluorogenic probes.

The combined results from all these experiments demonstrate that the Δ*bcnA*, Δ*lcoA*, and Δ*psrA* mutant cells display high level of peroxidation at the cell envelope with a discrete spatial distribution, being in general more pronounced at the poles and septal regions.

## The absence of BcnA, LcoA, and PsrA is associated with increased amounts of anionic membrane phospholipids

The discrete accumulation of the fluorescent peroxidation probes at cell poles, and in some cells also at the sites of cell division, suggested that anionic phospholipids, such CL and PG, could be the main peroxidized lipid species under stress since both lipids tend to accumulate in membrane regions that have negative curvature [9,59]. CL is a large anionic glycerol phospholipid composed of 2 phosphatidic acids connected by glycerol, giving it a conical shape. We used 3′,6-dinonyl neamine tetratrifluoroacetate (diNn) as a biological probe to investigate if the loss of BcnA, LcoA, and PsrA results in changes in the amount of anionic phospholipids. diNn is an aminoglycoside derivative that interacts with anionic phospholipids leading to membrane permeabilization and depolarization [60–63]. Compared to the wild-type strain (MIC 75 ± 28 μg ml$^{-1}$), the MIC of diNn was 6- and 3-fold lower in the Δ*bcnA* (MIC 12.5 μg ml$^{-1}$; $p = 0.004$) and Δ*lcoA* (MIC = 26 μg ml$^{-1}$; $p = 0.003$) mutants, respectively. In contrast, Δ*psrA* had an MIC = 107 μg ml$^{-1}$ ($p > 0.05$), suggesting no differences relative to the wild-type strain. These results indicate that loss of BcnA and LcoA, but not PsrA, could increase the permeability of the bacterial cell to the antibiotic probe or alternative stimulate increased amounts of anionic phospholipids.

To assess the level and distribution of anionic phospholipids in the bacterial cells, we employed fluorescence microscopy using Acridine Orange 10-nonyl bromide (NAO), a fluorophore with high affinity to CL and PG [64,65]. We compared Δ*bcnA*, Δ*lcoA*, and Δ*psrA* mutants against the wild-type strain in 2 conditions untreated and treated with 0.75 MIC of norfloxacin (46 μg ml$^{-1}$ for the wild-type strain and Δ*psrA* and 6 μg ml$^{-1}$ for Δ*bcnA* and Δ*lcoA*). Significantly higher level of fluorescence was seen in the mutant *bcnA* cells compared to the wild type (Fig 4A). K56-2 treated with norfloxacin showed a similar amount of anionic phospholipid as untreated Δ*bcnA* (Fig 4A). Although Δ*lcoA* and Δ*psrA* show higher levels of intensity when exposed to antibiotic stress, these mutants show almost identical levels of fluorescence to the wild type in the untreated condition (Fig 4A). Comparably to what was found with NBD-Pen and (H$_4$BPMHC), we observed higher fluorescence at the poles of all strains under antibiotic stress. Image quantification revealed that the fluorescence intensity in Δ*bcnA* bacterial cells was higher than that obtained in the wild type ($p = 0.0001$), while Δ*lcoA* and Δ*psrA* exhibited no significant differences with the wild-type strain in the untreated condition ($p > 0.05$; Fig 4B). However, under antibiotic stress, these mutants evidence a significant increase in their fluorescence than the wild-type strain treated with norfloxacin ($p = 0.0001$; Fig 4B). Similar results were also confirmed under Polymyxin B stress (S10 Fig). Together, these results suggest that the loss of the BcnA, LcoA, and PsrA proteins is associated with increased accumulation of anionic phospholipids at sites of maximal peroxidation under antibiotic stress.

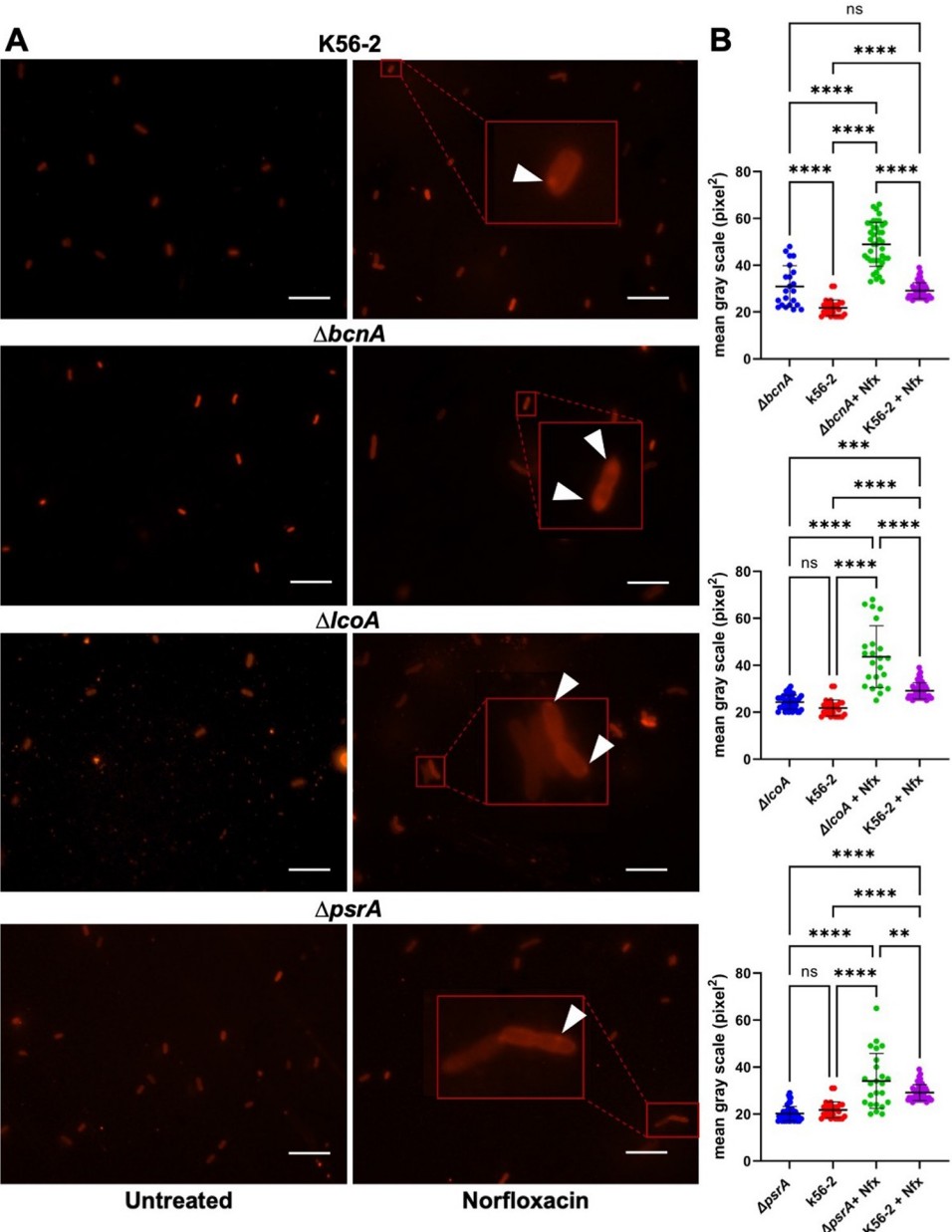

**Fig 4. Anionic phospholipid detection by fluorescence microscopy. (A)** Upon treatment with 1 μM NAO for 15 minutes at room temperature, bacteria were examined with a Zeiss microscope using a 100× oil immersion lens with an excitation at 640 nm. Arrowheads indicate the polar accumulation of fluorescence in bacterial cells poles. Scale bars, 10 μm. **(B)** Quantification of fluorescence intensity by ImageJ; the results represent the mean ± SD from 3 independent experiments. The statistical significance was determined by 1-way ANOVA with Tukey multiple comparisons test's post hoc analysis. [****], $p < 0.0001$; [***]; $p < 0.001$; ns, nonsignificant. Data underlying the graphs in this figure can be found in S2 Data. NAO, Acridine Orange 10-nonyl bromide; SD, standard deviation.

We also employed quantitative lipidomics to determine the overall lipid content in wild type and the Δ*bcnA*, Δ*lcoA*, and Δ*psrA* mutants using mass spectrometry. We cultivated mutants and wild type in the absence and in the presence of 1,024 μg ml⁻¹ polymyxin B. No significant changes in the amount of PE, a zwitterionic phospholipid that makes approximately 70% of the membrane in Gram-negative bacteria [66] were observed by comparing in K56-2

and the mutants, both in the absence or presence of antibiotic stress (S11A and S11B Fig, furthest left panels). Also, PG and CL showed no differences between mutant and wild-type strains when grown under no antibiotic stress (S11A Fig, central left panel and central right panel, respectively). On the contrary, 2′ alanyl-phosphatidylglycerol (APG), a low abundance PG derivative [67], showed a lower ratio in the mutants compared to the wild-type strain (S11A Fig, furthest right panel). Together, these results show that without antibiotic stress the mutants do not display significant changes in the 3 most abundant membrane phospholipids when compared to the *B. cenocepacia* K56-2 wild-type strain.

In contrast, under polymyxin B-induced antibiotic stress, Δ*bcnA* produced more PG than wild type, while Δ*psrA* and Δ*lcoA* showed similar quantities as in the wild-type strain (S11B Fig, central left panel). Also, Δ*bcnA* and the wild-type strain produced increased levels of CL under antibiotic stress in comparison to Δ*psrA* and Δ*lcoA* (S11B Fig, central right panel). Like in the absence of polymyxin B, mutants produced less APG than wild type under antibiotic stress (S11B Fig, furthest right panel). Finally, the PE phospholipid levels show no differences between strains in the same way as in the condition without polymyxin B (S11B Fig, furthest left panel). These results suggest that antibiotic stress and mutations in these genes produce modifications in the anionic phospholipids (PG and CL), consistent with the results using diNn and NAO.

## Discussion

This study has identified that *B. cenocepacia* mutants lacking the ability to produce BcnA, LcoA, and PsrA, 3 proteins widely conserved in many bacterial genomes, display enhanced membrane lipid peroxidation under conditions that provoke peroxidative stress, suggesting that they are components of a previously unrecognized detoxification system. Lipocalins are an ancient family of proteins found in all kingdoms of life; they can bind hydrophobic ligands but exhibit different functions depending on the cell types and organisms [28,68]. Their function in bacteria is unclear [69]. The first described bacterial lipocalin was the *E. coli* Blc, an outer membrane lipoprotein with high in vitro affinity for fatty acids and phospholipids, which was proposed to play a role in phospholipid transport to the outer membrane [70]. However, Blc orthologs are restricted to fewer bacteria, while BcnA orthologs are widespread in most Gram-positive and Gram-negative bacterial species. Previously, we discovered that *B. cenocepacia* BcnA and several of its orthologs contribute to antibiotic resistance by scavenging bactericidal antibiotics in the bacterial milieu [27,29]. We also suggested that while antibiotic scavenging could enhance intrinsic resistance to multiple antibiotics, the physiological function of the bacterial lipocalins could involve an antioxidant mechanism [27]. Supporting this notion, we demonstrate here that the absence of BcnA is associated with increased membrane peroxidation, increased production of peroxidation aldehyde by-products, and reduced bacterial survival in the cold. A role for lipocalins in protection against lipid peroxidation from cold and oxidative stress has been documented in plants, such as *Arabidopsis thaliana* [43] and *Medicago falcata* [71]. Further, the mammalian odorant–binding proteins, which are soluble lipocalins produced by the nasal mucosa in several mammalian species, can scavenge the peroxidized lipid by-product 4-hydroxyl-2-nonenal [72] and protect *E. coli* from peroxide-induced stress [73]. The human tear lipocalin (also known as lipocalin-1) can also act as a scavenger of lipid peroxidation products [74]. Therefore, scavenging toxic lipid peroxidation products may be a common feature of lipocalins in animals, plants, and bacteria.

BcnA could also exert a protective effect by detoxifying reactive toxic aldehydes or reactive oxidation intermediates in the periplasmic space. The mammalian lipocalin α-1 microglobulin has reductase and dehydrogenase activities related to a free cysteine group located in a flexible

loop [75]. However, BcnA and its homologs lack cysteines, precluding a similar mechanism to that of α-1 microglobulin. The *N. meningitidis* BcnA homolog (GNA1030) strongly binds Ubi-8 [37], a prenylated benzoquinone whereby the quinone moiety is a cofactor with antioxidant properties [36]. Donnarumma and colleagues [37] proposed that GNA1030 is an Ubi-8 transporter involved in respiration and electron transfer reactions, but the results using inhibitors were inconclusive. Conceivably, the reduced form (ubiquinol) of Ubi-8 quinone moiety could quench reactive oxidation intermediates, but this has not been directly examined.

The second protein identified in this study, LcoA, is a predicted cytochrome $b_{561}$ that could act synergistically with BcnA by reducing the oxidized quinone and restoring BcnA's reducing capacity, or independently, by directly reducing toxic reactive aldehydes in a manner analogous to the reduction of ascorbic acid as shown for its eukaryotic homologs [76]. Cytochrome $b_{561}$ proteins are transmembrane proteins that contain 2 *bis*-histidine coordinated hemes packed within the hydrophobic transmembrane helical bundles, which are located near opposing sides of the membrane interface and are part of electrogenic electron transfer systems. The cytochrome $b_{561}$ protein family in plants and animal cells are involved in electron transfer reactions using ascorbate as an electron donor while others act as $Fe^{3+}$ reductases [76]. Recently, a cytochrome $b_{561}$ ortholog in *Clostridioides difficile* was reported to play a role in protection against oxidative damage caused by inflammatory mediators and antibiotics [77], while the *E. coli* CybB cytochrome $b_{561}$ protein is a superoxide:ubiquinone oxidoreductase that directly oxidizes superoxide to molecular oxygen with the sequestered electron funneled to ubiquinone [78]. This function is important to scavenge superoxide produced by redox centers at intramembranous sites. Presumably, LcoA could function by reducing an oxidized form of the Ubi-8 bound to BcnA, which, in turn, can be reoxidized by peroxyl radicals arising from the oxidation of membrane lipids, thus providing a mechanism to quench peroxidation radicals in the periplasmic space. This model is consistent with the strong conservation of *lcoA* and *bcnA* within the same locus in many Gram-negative bacteria, including cases in which both genes are fused. Periplasmic membrane lipid–derived oxidative aldehydes escaping quenching by BcnA/LcoA could accumulate in the cytosol and be ultimately detoxified by cytoplasmic aldehyde reductases such as PsrA. This could explain why MDA accumulates in the Δ*psrA* mutant and leads to a concomitant increase in membrane lipid peroxidation, as we have observed in this work.

Our experiments imaging lipid peroxyl radicals in vivo with 2 fluorescent probes revealed a polar and midcell location of most of the fluorescence signals. The lipids in bacteria are not polyunsaturated and therefore less susceptible to peroxidation damage. The polar and midcell location of the peroxidation suggested that CL and PG could be the main peroxidized lipid species under stress, since they predominate at membrane regions that have negative curvature, such as the poles and septa of Gram-negative rods [9,59]. This was supported by 3 observations. First, Δ*bcnA* and Δ*lcoA* exhibited increased sensitivity to diNn, an aminoglycoside derivative that has been shown to interact with anionic phospholipids [60]. Second, Δ*bcnA* showed increased amount of PG under antibiotic stress conditions. Third, NAO excimer fluorescence, a reagent that preferentially binds anionic phospholipids, revealed greater fluorescence intensity in the Δ*bcnA*, Δ*lcoA*, and Δ*psrA* mutants under antibiotic stress. These observations, together with the absence of quantitative differences in the amounts of PE in the in the Δ*bcnA*, Δ*lcoA*, and Δ*psrA* mutants, suggest that anionic phospholipids are the primary targets of peroxidation, especially when localized to the bacterial poles and septal regions. CL is a minor anionic phospholipid that accumulates in the septum and poles of bacterial rods. However, quantitative differences in the CL content of Δ*bcnA*, Δ*lcoA*, and Δ*psrA* mutants both under steady state and after antibiotic challenge were not observed. This suggests that peroxidation stress may be associated with a redistribution of membrane lipids, which would explain

the discrete localization of anionic phospholipids at the same sites (poles and septa) where maximal peroxidation is detected without dramatic changes in overall lipid amounts. Further studies are underway to directly determine the oxidation state of the membrane lipids in the $\Delta bcnA$, $\Delta lcoA$, and $\Delta psrA$ mutants when they are placed under chemical or physical oxidative stress.

In summary, this work reveals a hitherto unidentified participation of the *B. cenocepacia* BcnA, LcoA, and PsrA proteins in detoxifying bacterial cells upon stress conditions that provoke membrane lipid peroxidation. The conservation of these proteins across many bacterial genera suggests they could provide a general bacterial protection mechanism. Further studies are required to elucidate the molecular mechanism of BcnA, LcoA, and PsrA action and to identify additional protein components involved in the control of peroxidation in the bacterial cell envelope.

## Materials and methods

### Reagents, bacterial strains, and plasmids

The reagents used in this study and their source are listed in S1 Table. Antibiotics were diluted in water, except for polymyxin B, which was diluted in 0.2% (w/v) bovine serum albumin with 0.01% (v/v) acetic acid. The bacterial strains and plasmids used in this study are listed in S2 Table. Bacteria were cultured in high-salt Luria-Bertani (LB) broth (Melford Biolaboratories, Chelsworth, UK) at 37˚C. All freezer stocks were maintained in LB broth + 20% glycerol at −80˚C. *B. cenocepacia* and *E. coli* strains were grown in LB broth at 37˚C and supplemented with antibiotics as necessary (100 μg ml$^{-1}$ trimethoprim, 50 μg ml$^{-1}$ gentamicin, and 150 μg ml$^{-1}$ tetracycline, for *B. cenocepacia* strains; 40 μg ml$^{-1}$ kanamycin and 30 μg ml$^{-1}$ tetracycline for *E. coli* strains). *E. coli* GT115 and DH5α competent cells were transformed via heat-shock/ calcium chloride treatment, as previously described [79]. Transformants were plated on LB agar plates with appropriate antibiotic selection. For microscopy, strains were inoculated in LB broth from colonies and grown at 37˚C with shaking until mid- to late-exponential phase, corresponding to an optical density at 600 nm ($OD_{600}$) of 1.0.

### Strain manipulations

MIC assays were carried out by microdilution. To determine the potassium tellurite MIC, the final ranges of drug dilutions were 3 to 100 μg ml$^{-1}$; for the diNn MIC assay, the final concentration range was 8 to 128 μg ml$^{-1}$. In both cases, inocula were prepared by diluting an overnight broth culture of each of the selected isolates in double-strength cation-adjusted (final concentration of 25 μg ml$^{-1}$ $CaCl_2 \cdot 2 H_2O$ + 10 μg ml$^{-1}$ $MgCl_2 \cdot 6 H_2O$ [80]) Mueller-Hinton broth (Sigma-Aldrich, Merck Life Sciences UK, Gillingham, UK) to reach a final inoculum of $5 \times 10^5$ CFU ml$^{-1}$. Bioscreen honeycomb microplates (Thermo Scientific, UK) were used for the MIC assays. Each well received 100 μl of the drug dilution and 100 μl of the inoculated double-strength broth. Water diluent was used as the no antibiotic control. Cultures were incubated at 37˚C with shaking in a Bioscreen C automated growth curve analyzer (Oy Growth Curves, Finland) for 20 hours. The MIC end point was read as the lowest drug concentration at which the % optical density at $OD_{600}$ relative to antibiotic-free control was ≤10% (which corresponded to no visible growth).

For bacterial growth under cold stress, overnight cultures of *B. cenocepacia* strains were diluted in LB broth to a final concentration of $10^3$ CFU ml$^{-1}$ and incubated at 8˚C for 24 hours. Samples were withdrawn and serially diluted in PBS, pH 7.4 (Gibco). A total of 10-μl portions were deposited onto the surface of LB agar plates. The plates were incubated at 37˚C for 24 hours, and the resulting colonies were counted.

For exposure to chemical oxidative stress, *B. cenocepacia* cultures in LB broth were challenged with 30 μM TBH with bacterial inoculum of $10^6$ CFU ml$^{-1}$ for 1 hour at 37˚C with shaking. After the treatment, samples were withdrawn and serially diluted in PBS. A total of 10-μl aliquots were spotted onto the surface of LB agar plates. The plates were incubated at 37˚C for 24 hours, and the resulting colonies were counted.

For growth under microaerophilic conditions bacterial cultures with an OD$_{600}$ of 0.00001 ($10^3$ CFU ml$^{-1}$) in LB broth were challenged with 0.25× MIC of norfloxacin for 4 hours [29] and incubated aerobically at 37˚C at 180 rpm. Samples were serially diluted in PBS. A total of 10-μl aliquots were spotted onto the surface of LB agar plates to enumerate colonies, and the plates were incubated at 37˚C for 24 hours aerobically or under microaerophilic conditions (5% $O_2$, 10% $CO_2$, and 85% $N_2$) [49] using the CampyGen Compact system (Oxoid, Basingstoke, UK) in anaerobic jars, as described by the manufacturer.

## Recombinant DNA methods

DNA amplification, restriction digests, and ligations were performed according to the manufacturer's instructions. Restriction enzymes and T4DNA ligase were from New England Biolabs, Hitchin, UK. Cloning steps were carried out in *E. coli* DH5α (S2 Table), and plasmids were purified using Plasmid DNA Miniprep kits (QIAGEN, Germantown, MD, USA). DNA sequencing was performed by Eurofins with double stranded plasmid or PCR templates, which were purified with QIAquick PCR purification kit (QIAGEN). Plasmids were introduced into *E. coli* strains by transformation, as described above. For colony PCR, 10 colonies were selected from each plate, and each colony was suspended in 50-μl sterile distilled water, heated at 100˚C for 10 minutes, and then centrifuged at 8,000 rpm for 4 minutes. Moreover, 2-μl aliquots of the supernatant were used as DNA template for PCR amplifications with Taq polymerase (QIAGEN). For cloning experiments, we used HotStar HiFidelity polymerase (QIAGEN). PCR amplifications were done according to the manufacturer's instructions and optimized for each primer pair.

## Construction of plasmids

A plasmid constitutively expressing LcoA for complementation was constructed by amplifying the *lcoA* coding sequence using the primers indicated in S3 Table. Amplicons containing the *lcoA* coding region were digested with *Nde*I-*Xba*I, purified with a QIAquick PCR purification kit (QIAGEN), and ligated into plasmid pDA17, previously digested with *Nde*I-*Xba*I. Resulting colonies after electroporation in *E. coli* DH5a were examined for colony PCR (as described above) using primers (S3 Table) that anneal with the flanking sequences of the multiple cloning sites of the pDA17 cloning vector. One of the PCR positive colonies was selected and the plasmid, named pDA17-LcoA was validated by DNA sequencing.

For the construction of an unmarked nonpolar deletion mutant of *psrA*, mutagenic plasmids were constructed as described using the method of Flannagan and colleagues [81,82]. Approximately 500-bp DNA fragments flanking both the upstream and downstream regions of the gene were PCR amplified using the listed primers in S3 Table and cloned into the suicide vector pGPI-SceI [81] to generate pDel*psrA* (S2 Table).

## Deletion mutagenesis

Mobilization of plasmids into *B. cenocepacia* was conducted by triparental mating [83]. *E. coli* GT115 carrying the mutagenic plasmid (Donor strain) was grown in LB broth supplemented with 50 μg ml$^{-1}$ trimethoprim; *E. coli* DH5α carrying pRK2013 (Helper strain [84]) was grown in LB broth supplemented with 40 μg ml$^{-1}$ of kanamycin. The recipient bacteria (*B.*

*cenocepacia*) were also grown in LB broth. Overnight cultures at 37˚C of the 3 strains were mixed at a ratio donor:helper:recipient (based on $OD_{600}$) of 3:3:1. Moreover, 1 ml of donor strain, 1 ml of helper strain, and 330 µl of recipient strain were sedimented separately by centrifugation at 4,000 rpm for 2 minutes. Cell pellets were combined and resuspended in 1 ml of LB broth, washed twice with LB broth, resuspended in 100 µl SOB broth (2% tryptone, 0.5% yeast extract, 0.05% sodium chloride, 0.24% magnesium sulfate, and 0.0186% potassium chloride), spotted onto a SOB agar plate, and incubated right face up at 37˚C overnight. The next day, the bacterial spot was scraped and resuspend in 1 ml of sterile LB broth 100 µl of the undiluted suspension and $10^{-1}$, $10^{-2}$, and $10^{-3}$ dilutions were deposited on LB agar plates supplemented with 100 µg $ml^{-1}$ trimethoprim (to select for *Burkholderia* cointegrants) and 50 µg $ml^{-1}$ gentamicin (to kill the helper and donor *E. coli* strains). The plates were incubated at 37˚C for 2 days. The next step was to generate a double crossover mutant using the pRK2013 helper plasmid and donor *E. coli* DH5α pDAI-SceI-SacB [82]. *E. coli* carrying pDAI-SceI-SacB was grown in LB broth with 30 µg $ml^{-1}$ tetracycline and *E. coli* carrying pRK2013 grew in LB broth supplemented with 40 µg $ml^{-1}$ kanamycin. Moreover, 3 to 5 colonies of the recipient bacteria (*Burkholderia* plus integrated suicide vector) were resuspended in 1 ml LB broth with 100 µg $ml^{-1}$ trimethoprim. Triparental mating was conducted as described above and exconjugants selected on LB agar containing 150 µg $ml^{-1}$ tetracycline (to select for *Burkholderia* carrying pDAISceI-SacB) and 50 µg $ml^{-1}$ gentamicin (to kill the helper *E. coli*) upon incubation for 2 days at 37˚C. A total of 20 different colonies were patched onto 2 LB agar plates, one containing 150 µg $ml^{-1}$ tetracycline and 50 µg $ml^{-1}$ gentamicin and the other containing 100 µg $ml^{-1}$ trimethoprim [82]. The pDAI-SceI-SacB plasmid was cured by growing the deletion mutants in LB broth without antibiotics for 24 hours. Then, 50 µl of each dilution were plated on 5% (w/v) sucrose LB agar plates lacking salt and incubated at 37˚C overnight. The resulting colonies were patched onto LB agar and LB agar plus 150 µg $ml^{-1}$ tetracycline; tetracycline sensitivity indicated loss of pDAI-SceI-SacB [82]. The mutants obtained were reisolated and stored at −80˚C.

## Determination of membrane lipid peroxidation levels

Membrane lipid peroxidation was assessed by the DPPP method as previously described with few modifications [41]. DPPP (Invitrogen, ThermoFisher, Paisley, UK) was dissolved in DMSO (Sigma-Aldrich) to prepare a 1mM stock solution; vials were flashed with nitrogen and stored in the dark at −20˚C until use. Overnight cultures in LB broth were washed twice and diluted to an $OD_{600}$ of 1 in PBS containing 100 µM DPPP and incubated for 60 minutes at room temperature with MIC levels of potassium tellurite in the dark. Fluorescence from DPPP = O was measured with a POLARstar Omega microplate reader (BMG LABTECH, Ortenberg, Germany) using 351 nm wavelength for excitation and 420 nm for emission.

## Determination of malondialdehyde levels

Overnight cultures were diluted to $OD_{600}$ of 0.1 and growth until reaching an $OD_{600}$ of 0.5 to 0.6 in 50 ml LB broth. Cells were treated with 0.5 MIC of norfloxacin [29] or 1,024 µg $ml^{-1}$ polymyxin B for 2 hours and incubated at 37˚C with shaking at 180 rpm. Bacteria were harvested, resuspended in PBS containing 1x protease inhibitor (cOmplete ULTRA Tablets, Mini, *EASYpack* Protease Inhibitor Cocktail, Merck Life Sciences UK, Gillingham, UK) and 1 µg $ml^{-1}$ DNase I (Roche VWR International, Lutterworth, UK), and were lysed using a cell disruptor at 27 kpsi (Constant Systems, Daventry, UK). Cell debris was removed after centrifugation at $10,000 \times g$ for 15 minutes at 4˚C. Samples were cleared by centrifugation at $16,000 \times g$ for 60 minutes at 4˚C (Sorvall RC 6 Plus, Germany), and the supernatant (total soluble proteins) was frozen at −80˚C. The Bradford assay was used to quantify protein

concentration. The ELISA assay for MDA quantification was performed as previously described [3] with few modifications. Moreover, 4 μg of sample proteins were diluted into 1 ml of coating buffer (carbonate/bicarbonate, 100 mM, pH 9.6). A total of 200 μl of each sample was added to ELISA plates (Nunc MaxiSorp ThermoFisher, Paisley, UK). Control wells contained only coating buffer. The plate was covered with sealing tape and incubated at 4°C overnight. After the incubation, the plate was washed 5 times with 300 μl of PBS/Tween20 (0.05%) per well. Furthermore, 250 μl of 5% BSA (blocking buffer) in PBS, pH 7.4, was added to each well and incubated at room temperature for 2 hours. Plates were washed as above and 200 μl of 1:500 anti-MDA antibody (Abcam, UK) was added and incubated at 37°C for 1 hour. Plates were washed 4 times with PBS/Tween20, and then a secondary goat anti-rabbit horseradish peroxide (HRP, Abcam) antibody (1:10,000) was added and incubated for 1 hour at room temperature. This was followed by a final wash, and then 200 μl of 3,3′,5,5′-tetramethylbenzidine was added to each well and incubated in the dark at room temperature. After enough color development, 100 μl of 2 M HCl was added. Absorbance of each well was measured with POLARstar Omega microplate reader (BMG LABTECH) at 570 nm.

## Analysis and quantification membrane lipids by mass spectrometry

Bacterial samples were taken from LB broth in biological triplicates. Cells were centrifuged at $3,100 \times g$ at 4°C and normalized to an $OD_{600}$ of 0.5. Lipids were extracted from the bacterial pellets via a modified Folch extraction method as detailed previously [85]. Before extraction, a final concentration of 500 nM of lipid standard, sphingosyl phosphatidylethanolamine (SPE; d17:1/12:0 sphingosyl PE, Avanti Polar Lipids, Birmingham, AL, USA), was added to each sample. The pellets were resuspended by vortexing with 500 μl of ice-cold LC-MS grade methanol. Moreover, 300 μl of ice-cold sterile water was added before 1 ml of ice-cold HPLC grade chloroform. Each sample was vortexed for 30 seconds before centrifugation at $1,739 \times g$ for 15 minutes at 4°C. The lower phase was extracted, and the solvent from this phase was removed under a stream of nitrogen gas. Dried lipids were resuspended in 95:5 acetonitrile:ammonium acetate and methanol at a 2:1 ratio, respectively. Samples were analyzed by LC/MS using a Dionex Ultimate 3000LC/MS (amaZon SL, Bruker Daltonics, Bruker, Banner La, UK) as shown previously [84]. Lipids were separated by HPLC using a BEH Amide XP column (Waters, Wilmslow, UK). The column was maintained at 30°C with a flow rate of 150 μl/min. Each run consisted of a 10-minute calibration with 95% acetonitrile and 5% 10 mM ammonium acetate (pH 9.2) followed by injection of 5 μl of the sample. The sample was run on a 15-minute gradient of 95% to 70% acetonitrile. Each sample was analyzed in positive and negative ionization modes. The 4 major phospholipids (PE, PG, CL, and APG) were quantified in QuantAnalysis (Bruker Daltonics) as shown previously [84], which calculates the area under each peak for each sample. The sum of these areas was compared to that of the known lipid standard, SPE.

## Fluorogenic detection of lipid peroxidation

For experiments using NBD-Pen, overnight bacterial cultures were diluted and grown in 5-ml LB until the $OD_{600}$ reached 0.5 to 0.6. Then, bacteria were challenged with sub-MIC of norfloxacin for 4 hours (K56-2 MIC- 64 μg ml$^{-1}$, ΔbcnA MIC- 8 μg ml$^{-1}$ [29]) or 100 μg ml$^{-1}$ of polymyxin B. The cells were centrifuged and washed 3 times with PBS. Next, 10 μl of bacteria was transferred into a new Eppendorf tube and 10 μl of 2 μM NBD-Pen was added. The tubes were incubated in the dark at room temperature for 15 minutes, as NBD-Pen is light sensitive and rapidly oxidizes. Then the bacterial cells were centrifuged and washed 3 times with phosphate buffered saline. Moreover, 5-μl samples were deposited onto a 0.8% agarose coated microscopic slide and covered with a coverslip. The samples were examined using a Zeiss

microscope using a 100× oil immersion lens and using GFP filter for NBD-Pen excitation wavelength ($\lambda$ex) of 470 nm [50].

For experiments using the fluorogenic $\alpha$-tocopherol analogue, overnight bacterial cultures were diluted to an $OD_{600}$ of 1 in 900 $\mu$l LB, stained with 100 $\mu$l of 10 $\mu$M H$_4$BPMHC [57] for 15 minutes in the dark. Then the bacterial cells were centrifuged and washed 3 times with PBS. Moreover, 5-$\mu$l samples were deposited onto 0.8% agarose-coated microscopic slides and covered with a coverslip. The samples were investigated using a Zeiss microscope using a 100× oil immersion lens and using GFP filter for H$_4$BPMHC with $\lambda$ex of 488 nm [56–58].

## Fluorogenic detection of anionic phospholipids

Overnight bacterial cultures were diluted to an $OD_{600}$ of 0.1 in LB and grown until an $OD_{600}$ of 0.6 in 2 conditions. Untreated or treated with 0.75 MIC of norfloxacin or 100 $\mu$g ml$^{-1}$ of polymyxin B. After that, bacterial cells were washed 3 times with PBS, stained with 1 $\mu$M fluorescent dye NAO (Invitrogen) for 15 minutes in the dark. A total of 10-$\mu$l samples were deposited onto a 0.8% agarose coated microscopic slide and covered with a coverslip and examined using a Zeiss microscope using a 100× oil immersion lens and the appropriate filter for NAO ($\lambda$ex = 488 nm/$\lambda$em = 639 nm) [60]. The incorporation of NAO into membranes was detected as red fluorescence emitted by the formation of NAO excimers [60,65].

## Image analysis

Most imaging experiments were performed independently by 2 individuals, and the quantitation was performed with multiple images to reduce selection bias. Quantification of bacterial peroxidation and CL by fluorogenic probes was performed using Fiji Is Just ImageJ (FIJI) [86] as previously described by Christine Labno (https://www.unige.ch/medecine/bioimaging/files/1914/1208/6000/Quantification.pdf). The analysis was based on determining the average gray values of each bacterial cell, where the sum of gray values is divided by the number of pixels. Briefly, micrographs were imported into FIJI and converted into 8-bit images and subsequently transformed into binary images. If required, background noise was removed with the FIJI "Despeckle" function. Using the measurement tool, mean gray values were measured to determine the fluorescence intensity, and the results were automatically exported into a cvs file and processed with GraphPad Prism version 9, GraphPad Software, San Diego, CA, USA to prepare graphs and perform statistical analysis. Only for the preparation of figures, contrast of each file was globally enhanced with the FIJI "Enhance contrast" using 0.0005%, followed by application of the Photoshop (version 20.0.1 Adobe, San Jose, CA, USA) unsharp mask (150%, pixel radius 2 and threshold 3). Insets in images were added using the ImageJ Zoom_in_Images_and_Stacks macro (https://imagej.nih.gov/ij/macros/tools/Zoom_in_Images_and_Stacks.txt) with a Zoom factor of 4.

## Quantification of the fluorescence intensity across the long axis of the cell

Radial distribution of the average fluorescent intensity, of the NBD-Pen dye, across the long axis of the cell, was quantified using the MicrobeJ [87] plug-in for Fiji [86]. Briefly, phase contrast and fluorescent images were stacked together using the stack option in MicrobeJ, and bacteria were identified in the phase contrast channel using the bacterial detection option (in the "bacteria" tab of MicrobeJ). This allows the identification of bacteria in the fluorescent channel. To calculate the subcellular localization of the fluorescent intensity, we used the intensity detection analysis in the "maxima" tab and selected the association option to associate the intensity to each of the previously identified bacterial cells in the image. The fluorescence intensity across the long axis of the cell was normalized to the average intensity of the cell under no treatment. The average intensity was obtained using MicrobeJ. The results were

saved to a cvs file. The fluorescence intensity distribution and the identification of intensity peaks were graphed using Python version 3.8 (http://www.python.org) script, which can be accessed in GitHub (https://github.com/jdYEXzhM/Quantification/blob/main/Quantification.py).

## Bioinformatics

The synteny analysis to evaluate the genetic neighborhood of *bcnA* homologs in different β-proteobacteria and other bacterial families was based on MultiGeneBlast [39]. Genomes were downloaded from NCBI, https://www.ncbi.nlm.nih.gov, to produce a database, following the instruction for MultiGeneBlast. For the architecture search, the query consisted of YceI lipocalin family sequences downloaded from the pfam database, http://pfam.xfam.org. The database was searched for significant hits applying, 20% minimal identity of Blast hit for the query, and 20 kb as the maximum kilobase distance between genes. All the other parameters were set as default.

## Three-dimensional modeling of PsrA

The amino acid sequence of the putative aldehyde reductase from *B. cenocepacia*, PsrA, was downloaded from https://burkholderia.com. The crystallized structure of the YqhD aldehyde reductase of *E. coli* was downloaded from RCSB PDB (https://www.rcsb.org) using the PDB ID: 1OJ7 [88]. The three-dimensional modeling of the PsrA structure was made using the I-TASSER online server (https://zhanglab.dcmb.med.umich.edu/I-TASSER) [89] using the crystalized structure of *E. coli* as a template for modeling. The crystallized structure of *E. coli* was used as the template for generating the predicted model. The structural comparison analysis was made using the align function of PyMOL (The PyMOL Molecular Graphics System, Version 2.0, Schrödinger. Cambridge, UK).

## Statistical analysis

Statistical analyses were conducted with GraphPad Prism 9.0. The results were expressed as mean ± standard deviation (SD). Statistics were performed using 1- or 2-way ANOVA and *t* test, as required. Experiments were conducted using a minimum of 3 biological repeats, each with a minimum of 2 technical repeats. Additional statistical information is provided in the relevant figure legends.

## Supporting information

**S1 Fig. Synteny analysis of *lcoA*, *bcnA*, and *bcnB* gene architecture in members of the proteobacteria.** The analysis was performed by MultiGeneBlast as described in Materials and methods. Genes are not drawn to scale. The β-proteobacteria members in which *bcnA* is monocistronic are indicated in cyan.
(TIF)

**S2 Fig. Response of wild-type and mutant strains to *tert*-butyl-hydroperoxide.** Bacterial cultures were challenged with either 30 μM (final concentration) *tert*-butyl-hydroperoxide (TBH+) or vehicle control (TBH−) for 1 hour at 37°C. Results are shown as the mean CFU/ml ± SD of surviving bacteria from 3 independent biological replicates in duplicate. *p*-Values relative to the CFU/ml recovered from the wild-type strain K56-2 were calculated by 2-way ANOVA with the Sidak multiple comparison test. $^{*}$, $p \leq 0.05$; $^{**}$, $p \leq 0.01$ $^{***}$, $p \leq 0.0001$. When significant, the calculated *p*-values are shown. Data underlying the graph in this figure can be found in S1 Data. CFU, colony-forming unit; SD, standard deviation; TBH, tert-butyl hydroperoxide.
(TIF)

**S3 Fig. Antibiotic stress-induced accumulation of MDA in the absence of BcnA, LcoA, and PsrA, and oxygen dependent antibiotic susceptibility.** K56-2 and mutants carried the control plasmid vector pDA17 or the indicated complementing plasmid (in parentheses). **(A)** Bacteria were grown overnight with or without Nfx at concentrations corresponding to the 0.5 MIC (32 μg ml$^{-1}$ for the wild-type strain K56-2, Δ*bcnB*, Δ*lcoA*(pLcoA), and Δ*psrA*, 24 μg ml$^{-1}$ for Δ*bcnA* (pBcnA), 16 μg ml$^{-1}$ for Δ*lcoA*, 4 μg ml$^{-1}$ for Δ*bcnA*, and Δ*bcnAB*Δ*lcoA*). Results are shown as the mean of absorbance at 570 nm ± SD. Data represent the results of 3 independent experiments, each done in duplicate. The statistical significance of MDA levels untreated bacteria was determined by 2-way ANOVA. Strains comparisons were made against the wild-type K56-2 by Dunnett post hoc analysis. **(B)** Bacterial cultures were diluted and plated on LB plates with or without Nfx (at 0.25 MIC) and incubated in anaerobic jars under microaerophilic conditions. CFU were enumerated after 24-hour incubation at 37°C. Results are shown as the mean CFU/ml ± SD from 2 independent biological replicates in duplicate. **(C)** Control experiment as in (B), but under standard aerobic conditions. Results are shown as the mean CFU/ml ± SD from 2 independent biological replicates in duplicate. The statistical significance of results in panels A–C was determined by 2-way ANOVA. Individual comparisons were made against K56-2 by Dunnett post hoc analysis. *, $p \leq 0.05$, **; $p \leq 0.01$; ***, $p \leq 0.001$; ****, $p \leq 0.0001$; ns, nonsignificant. Data underlying the graphs in this figure can be found in S1 Data. CFU, colony-forming unit; MDA, malondialdehyde; MIC, minimum inhibitory concentration; Nfx, norfloxacin; SD, standard deviation. (TIF)

**S4 Fig. PmB stress-induced accumulation of MDA in the absence of BcnA, LcoA, and PsrA.** Wild-type strains and mutants contained the control plasmid vector pDA17 or the indicated complementing plasmid (in parentheses). Bacteria were grown overnight with or without 1,024 μg ml$^{-1}$ PmB. MDA was determined by ELISA (see Materials and methods). Results are shown as the mean of absorbance at 570 nm ± SD. Data represent the results of 3 independent experiments, each done in duplicate. The statistical significance of MDA levels obtained from PmB-treated versus untreated bacteria was determined by 2-way ANOVA. Strains comparisons were made against the wild-type K56-2 by Dunnett multiple comparison test. ****, $p \leq 0.0001$; ns, nonsignificant. Data underlying the graph in this figure can be found in S1 Data. MDA, malondialdehyde; PmB, polymyxin B; SD, standard deviation. (TIF)

**S5 Fig. Detection of lipid peroxidation in Δ*lcoA* and Δ*psrA* by fluorescence microscopy using NBD-Pen. (A)** Fluorescence (left) and phase-contrast (right) images of Δ*lcoA* and Δ*psrA* strains untreated or treated with 75% MIC for Nfx and incubated with NBD-Pen. Scale bars, 10 μm. **(B)** Quantitative analysis of fluorescence intensity based on fluorescence images in panel A. SD is shown as error bars, and mean values are shown as horizontal black lines. Statistical significance was determined by 1-way ANOVA with Tukey multiple comparisons test's post hoc analysis. ****, $p < 0.0001$; **, $p < 0.005$; ns, nonsignificant. Data underlying the graph in this figure can be found in S2 Data. NBD-Pen, 2,2,6-trimethyl-4-(4-nitrobenzo [1,2,5]oxadiazol-7-ylamino)-6-pentylpiperidine-1-oxyl; Nfx, norfloxacin; SD, standard deviation. (TIF)

**S6 Fig. Detection of lipid peroxidation by fluorescence microscopy using NBD-Pen. (A)** Fluorescence (left) and phase-contrast (right) images of K56-2 and Δ*bcnA* strains untreated or treated with 100 μg ml$^{-1}$ of PmB and incubated with NBD-Pen. Scale bars, 10 μm. **(B)** Quantitative analysis based on fluorescence images in panel A. SD is shown as error bars, and mean values are shown as horizontal black lines. Statistical significance was determined by 1-way

ANOVA with Tukey multiple comparisons test's post hoc analysis. ****, $p < 0.0001$; ***; $p < 0.001$; **, $p < 0.005$; ns, no significative. Data underlying the graph in this figure can be found in S2 Data. NBD-Pen, 2,2,6-trimethyl-4-(4-nitrobenzo [1,2,5]oxadiazol-7-ylamino)-6-pentylpiperidine-1-oxyl; PmB, polymyxin B; SD, standard deviation.
(TIF)

**S7 Fig. Detection of lipid peroxidation by fluorescence microscopy using NBD-Pen. (A)** Fluorescence (left) and phase-contrast (right) images of Δ*lcoA* and Δ*psrA* strains untreated or treated with 100 μg ml⁻¹ of PmB and incubated with NBD-Pen. Scale bars, 10 μm. **(B)** Quantitative analysis based on fluorescence images in panel A. SD is shown as error bars, and mean values are shown as horizontal black lines. ****, $p < 0.0001$, ns, no significative. Data underlying the graph in this figure can be found in S2 Data. NBD-Pen, 2,2,6-trimethyl-4-(4-nitrobenzo [1,2,5]oxadiazol-7-ylamino)-6-pentylpiperidine-1-oxyl; PmB, polymyxin B.
(TIF)

**S8 Fig. Distribution of peroxidation in the bacterial cell body.** Averaged green intensity profiles of cells labeled with NBD-Pen versus normalized cell length. The intensities along the long axis of each cell were averaged. The average value is depicted as a green line for each strain. The shaded space surrounding the NBD-Pen fluorescence intensity profiles designates the SD of the intensity at each point. The peak of fluorescent intensity produced in the treated bacteria is shown as green dots. Data underlying the graph in this figure can be found in S2 Data. NBD-Pen, 2,2,6-trimethyl-4-(4-nitrobenzo [1,2,5]oxadiazol-7-ylamino)-6-pentylpiperidine-1-oxyl; SD, standard deviation.
(TIF)

**S9 Fig. Detection of lipid peroxidation by fluorescence microscopy using H₄BPMHC. (A)** Overlay fluorescence and phase contrast images of bacterial cells incubated with 1 μM H₄BPMHC for 15 minutes at 37°C. The insets are zoomed sections of the images to provide more detail of the location of the fluorescent at the cell poles and mid cell (arrowheads). Scale bars, 5 μm. **(B)** Quantitative analysis of the fluorescent intensities of all bacterial cells; the results represent the mean ± SD from 3 independent experiments. Statistical significance was calculated by 1-way ANOVA with Tukey multiple comparisons test's post hoc analysis. ns, nonsignificant. Data underlying the graph in this figure can be found in S1 Data. H₄BPMHC, 8-((6-hydroxy-2,5,7,8-tetramethylchroman-2-yl)-methyl)-1,5-di(3-chloropropyl)-pyrromethene fluoroborate; SD, standard deviation.
(TIF)

**S10 Fig. Anionic phospholipid detection by fluorescence microscopy under PmB stress. (A)** Detection of CL using 1 μM NAO for 15 minutes at room temperature. Bacteria were examined with a Zeiss microscope using a 100× oil immersion lens with an excitation at 640 nm. Scale bars, 10 μm. **(B)** Quantification of fluorescence intensity by ImageJ; the results represent the mean ± SD from 3 independent experiments. The statistical significance of the relative mean grayscale levels obtained from treated (PmB) versus untreated bacteria was determined by 1-way ANOVA with Tukey multiple comparisons test's post hoc analysis. ****, $p < 0.0001$; ***; $p < 0.001$; **, $p < 0.005$; ns, nonsignificant. Data underlying the graph in this figure can be found in S2 Data. CL, cardiolipin; NAO, Acridine Orange 10-nonyl bromide; PmB, polymyxin B; SD, standard deviation.
(TIF)

**S11 Fig. Quantification on phospholipids in wild-type and Δ*bcnA*, Δ*lcoA*, and Δ*psrA* mutants.** CL, PG, PE, and 2′ APG were identified and quantified by mass spectrometry using

QuantAnalysis (Bruker Daltonics), which calculated the area under each peak for each sample. The sum of these areas was compared to that of the known lipid standard, SPE. **(A)** Lipids from bacteria under no antibiotic treatment. **(B)** Lipids from bacteria treated with 1,024 μg ml$^{-1}$ of PmB. SD is shown as error bars, and mean values are shown as bars. ****, $p < 0.0001$; ***; $p < 0.001$; **, $p < 0.005$; *, $p < 0.05$; ns, not significant. Data underlying the graphs in this figure can be found in S2 Data. APG, alanyl-phosphatidylglycerol; CL, cardiolipin; PE, phosphatidylethanolamine; PG, phosphatidylglycerol; PmB, polymyxin B; SD, standard deviation; SPE, sphingosyl phosphatidylethanolamine.
(TIF)

**S1 Table. Reagents.**
(DOCX)

**S2 Table. Bacterial strains and plasmids.**
(DOCX)

**S3 Table. Primers.**
(DOCX)

**S1 Data. Numerical values underlying Fig 1, S2 Fig, S3 Fig, S4 Fig and S9 Fig.**
(XLSX)

**S2 Data. Numerical values underlying Fig 2, Fig 3, Fig 4, S5 Fig, S6 Fig, S7 Fig, S10 Fig and S11 Fig.**
(XLSX)

## Author Contributions

**Conceptualization:** Marwa Naguib, Nicolás Feldman, Omar M. El-Halfawy, Miguel A. Valvano.

**Data curation:** Marwa Naguib, Nicolás Feldman, Paulina Zarodkiewicz, Holly Shropshire, Jean-Luc Décout, Miguel A. Valvano.

**Formal analysis:** Marwa Naguib, Nicolás Feldman, Paulina Zarodkiewicz, Holly Shropshire, Miguel A. Valvano.

**Funding acquisition:** Gonzalo Cosa, Miguel A. Valvano.

**Investigation:** Marwa Naguib, Nicolás Feldman, Paulina Zarodkiewicz, Holly Shropshire, Christina Biamis, Omar M. El-Halfawy, Julia McCain, Clément Dezanet, Jean-Luc Décout, Yin Chen, Gonzalo Cosa, Miguel A. Valvano.

**Methodology:** Marwa Naguib, Nicolás Feldman, Paulina Zarodkiewicz, Holly Shropshire, Christina Biamis, Omar M. El-Halfawy, Julia McCain, Clément Dezanet, Jean-Luc Décout, Yin Chen, Gonzalo Cosa.

**Project administration:** Miguel A. Valvano.

**Resources:** Holly Shropshire, Julia McCain, Clément Dezanet, Jean-Luc Décout, Yin Chen, Gonzalo Cosa, Miguel A. Valvano.

**Software:** Nicolás Feldman, Paulina Zarodkiewicz, Christina Biamis.

**Supervision:** Miguel A. Valvano.

**Validation:** Marwa Naguib, Nicolás Feldman, Paulina Zarodkiewicz, Holly Shropshire, Christina Biamis.

**Visualization:** Marwa Naguib, Nicolás Feldman, Paulina Zarodkiewicz, Holly Shropshire, Christina Biamis.

**Writing – original draft:** Marwa Naguib, Nicolás Feldman.

**Writing – review & editing:** Marwa Naguib, Nicolás Feldman, Paulina Zarodkiewicz, Holly Shropshire, Christina Biamis, Omar M. El-Halfawy, Julia McCain, Clément Dezanet, Jean-Luc Décout, Yin Chen, Gonzalo Cosa, Miguel A. Valvano.

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
