## [Editor Report · Decision Letter 0]

23 Nov 2021

Dear Dr. Valvano, 

Thank you for submitting your manuscript entitled "An evolutionary conserved detoxification system for membrane lipid-derived peroxyl radicals in Gram-negative bacteria" for consideration as a Research Article by PLOS Biology.

Your manuscript has now been evaluated by the PLOS Biology editorial staff and I am writing to let you know that we would like to send your submission out for external peer review. After our discussion we decided that we will consider this manuscript as a Discovery Report. Please select that option in our system when re-submitting your manuscript.

Discovery Reports describe novel and intriguing initial findings with the potential to lead to a significant new result for the field. Discovery Reports are short articles, typically with 2-4 main figures. While the research may be preliminary, studies should be advanced to the stage where observations or findings have been confirmed by independent methods or experimental approaches and obvious alternative interpretations have been ruled out. Discovery Reports are designed to work together with Update Articles to empower researchers to evaluate and share work in a way that more closely mirrors the real-world research process and create a comprehensive research story. 

Before we can send your manuscript to reviewers, we need you to complete your submission by providing the metadata that is required for full assessment. To this end, please login to Editorial Manager where you will find the paper in the 'Submissions Needing Revisions' folder on your homepage. Please click 'Revise Submission' from the Action Links and complete all additional questions in the submission questionnaire.

Once your full submission is complete, your paper will undergo a series of checks in preparation for peer review. Once your manuscript has passed the checks it will be sent out for review. 

If your manuscript has been previously reviewed at another journal, PLOS Biology is willing to work with those reviews in order to avoid re-starting the process. Submission of the previous reviews is entirely optional and our ability to use them effectively will depend on the willingness of the previous journal to confirm the content of the reports and share the reviewer identities. Please note that we reserve the right to invite additional reviewers if we consider that additional/independent reviewers are needed, although we aim to avoid this as far as possible. In our experience, working with previous reviews does save time. 

If you would like to send your previous reviewer reports to us, please specify this in the cover letter, mentioning the name of the previous journal and the manuscript ID the study was given, and include a point-by-point response to reviewers that details how you have or plan to address the reviewers' concerns. Please contact me at the email that can be found below my signature if you have questions. 

Please re-submit your manuscript within two working days, i.e. by Nov 25 2021 11:59PM.

Kind regards,

Paula

Paula Jauregui, PhD

Editor

PLOS Biology

---

## [Decision Letter · Decision Letter 1]

26 Jan 2022

Dear Dr Valvano,

Thank you for submitting your manuscript "An evolutionary conserved detoxification system for membrane lipid-derived peroxyl radicals in Gram-negative bacteria" for consideration as a Discovery Report at PLOS Biology. Your manuscript has been evaluated by the PLOS Biology editors, an Academic Editor with relevant expertise, and by one independent reviewer.

As you will see in the reviews attached below, the reviewer and the Academic Editor find your manuscript interesting and are overall positive about the experimental design. You will also note that the reviewer suggested a few additional experiments to validate the results and strengthen the work. After discussion with the Academic Editor whom we consulted for this decision, we recommend that the dose-response analyses suggested by reviewer 1 be treated as optional; these experiments would not be necessary for the current Discovery Report, but may instead be considered for inclusion in a follow-up Update Article in the future, if appropriate. The remainder of the reviewers' concerns, however, must be addressed.

Please also note that as your manuscript is being considered as a Discovery Report, we require that you reduce the total number of figures to four. You could either combine material from multiple figures or moved less crucial material into the supplementary figures.

In light of the reviews, we will not be able to accept the current version of the manuscript, but we would welcome re-submission of a revised version that takes into account the reviewer's and Academic Editor's comments, as well as our editorial requests. We cannot make any decision about publication until we have seen the revised manuscript and your response to the comments provided.

We expect to receive your revised manuscript within 1 month.

**IMPORTANT - SUBMITTING YOUR REVISION**

*Resubmission Checklist*

*Published Peer Review*

*PLOS Data Policy*

*Blot and Gel Data Policy*

Sincerely,

Dario

Dario Ummarino, PhD

Senior Editor

PLOS Biology

dummarino@plos.org

REVIEWS:

Reviewer #1: In their Discovery Report, Naguib and Feldman et al. report the characterization of BcnA, LcoA, and PsrA in Burkholderia cenocepacia as lipid peroxidation detoxifiers. Based on results from a combination of genetics, biochemical probes, microbiological assays, microscopy, and lipid analyses, the authors propose that all three proteins contribute to reducing membrane lipid peroxidation under diverse stresses (exogenous metal, antibiotics, and low temperature) by directly scavenging lipid peroxidation adducts. This Report is conceptually sound, and its findings may have applications to better understanding bacterial cell death and combatting antibiotic resistance. Additionally, this Report may certainly be of interest to PLOS Biology's microbiology audience, as well as other researchers who are interested in the role of reactive species in biology. I have only a few major suggestions, largely pertaining to validation of the results, that I believe would strengthen this work:

1. Dose-response: For all experiments with chemical stressors (potassium tellurite, TBH, norfloxacin, polymyxin B) described, treatment seems to be at only one dose/concentration. Potassium tellurite treatment is at MIC, and antibiotic treatment is at sub-MIC levels. However, MICs can vary by 100% even under identical experimental conditions. Additionally, if lipid peroxidation were indeed relevant to the actions of these chemical stressors, one might expect to find dose-response relationships, in which the observed readouts vary in relation to the dosage applied. I am therefore asking that the authors perform additional experiments where the dosage is varied (preferably including a dose at supra-MIC levels), and comment on whether a dose-response relationship emerges. Some places where this could be done in a straightforward manner are the fluorescence assays (in bulk culture and in single cells), the MDA assays, and the accompanying CFU/mL assays.

2. Relationship to lethality: In Fig. 2A, DPPP=O intensities for potassium tellurite treatment are presented, but there is no information shown about corresponding CFU/mL values under this treatment condition, as was done with antibiotics (Fig. 4). Similarly for TBH treatment in Fig. S1. The authors should, wherever applicable, measure the CFU/mL values and compare them to the fluorescence (and potentially other) readouts to better substantiate the claim that lipid peroxidation contributes to lethality.

3. Positive controls: The addition of positive ROS controls would facilitate comparison to known stressors. Exogenous addition of ROS such as H2O2 should lead to lipid peroxidation, which ought to be captured by the various probes the authors use. The authors might consider adding such a positive control to at least a few of the assays presented.

4. Lipidomics: In their lipidomics experiments, the authors find no significant changes in the amount of PE, but increased levels of CL and PG, as a result of polymyxin B treatment (Fig. 9). This appears partially similar to a previous observation in ref. 24, where the authors claimed larger CL/PE ratios after treatment with ciprofloxacin, kanamycin, and ampicillin in E. coli. However, in that work, PG does not seem to be significantly increased as a result of treatment. Could the authors better compare and contrast their lipidomics results to these and other results present in the literature?

Additional points:

1. Line 47: This line needs to be rephrased, as not all toxic reactive byproducts are reactive oxygen species (see, for instance, reactive nitrogen and electrophilic species).

2. Paragraph beginning on line 56: the focus on Gram-negatives is appropriate, but begs the question of what happens in Gram-positives. It would be helpful to readers if the authors could add a sentence or two mentioning this.

3. Line 72: I do not think "near-lethal" is correct, since at least one of the references (ref. 24) apply antibiotics at both sub-MIC and substantially lethal concentrations.

4. Line 148: Specifying the species again here would help the reader.

5. Line 194: Please name the gene before describing the knockout, or revise the presentation of this paragraph. The indicated knockout strain name was nowhere to be found in Fig. 2!

6. Line 232, the reference should be to Fig. 2B.

7. Line 290, since the authors only measure MDA, "byproducts" should not be plural.

8. Fig. 3, are the MDA levels normalized to bacterial protein concentrations? The methods mention a Bradford assay, but no normalization is mentioned in the figure legend. 

9. Fig. S5, presenting the figures normalized to mean untreated intensities as they vary along the cellular axis would make it easier for readers to compare the treatment effects.

10. Fig. 6, what is the rationale for changing the concentration to 75% MIC here? As mentioned above, the authors should perform dose-response experiments to confirm these results across a range of concentrations.

11. Figs. 5, 6, and 8 and associated SI figures need scale bars. Also, the units of pixel^2 seem incorrect. 

12. For visual clarity, Fig. 7 might be better presented as fluorescence and phase separately. The authors could add the overlay in addition to these two.

13. Some of the data for K56-2 in Figs. 6, S3, S4, and S6 seem inconsistent with those presented in Fig. 5. Are these the same data, and if not, why is the former mean greater than the latter mean? Additionally, some of the K56-2 data in Fig. 8 seem awfully similar to the K56-2 data in the other figures, but Fig. 8 uses a different fluorescence probe. Could the authors please clarify?

14. The CFU/mL ranges in Figs. 2 and 4 seem somewhat small, as they only span at most one log. These differences may be smaller than what some readers might be used to, for instance when examining treatment of other pathogens like E. coli and S. aureus with antibiotics (where differences in survival are typically at least one, if not multiple, logs). Perhaps the authors could comment on this difference if they feel it could improve the accessibility of their work. 

-------------

Comments from Academic Editor Bert van den Berg:

The paper by Valvano and colleagues describes a preliminary characterisation of the role of several gene products in overcoming lipid peroxidation. The authors present a number of experiments centered on the proteins BncA/B, LcoA and SprA in various strains of Burkholderia cenocepacia (including knockouts and complemented strains), which together paint a fairly convincing picture of their involvement in scavenging harmful reactive oxygen species that result from lipid peroxidation by environmental and chemical stressors, such as sublethal concentrations of antibiotics. While the current paper is probably overly long for a Discovery report (especially regarding the large number of figures), I do think it meets the criteria for consideration by PLOS Biology as the topic is interesting, novel, and provides many research questions that could be the focus of further work to confirm the present findings and gain mechanistic understanding. Overall, the manuscript is reasonably well-written and clear, but the fact that the main figure legends are separated from the figures themselves (and the same is true for the supplementary figures) made reviewing not as pleasant as it should have been. It would be appreciated if this were addressed in any revisions. Other than this I have the following questions and comments:

1. Since I'm not very familiar with the topic, I was wondering how important the prevention of lipid peroxidation is under "regular" conditions. It seemed to me that the chemicals and conditions used to induce the effects of BncA etc are somewhat exotic. Can the authors make a reasonable argument that the main role of these proteins is indeed to limit the harmful effects of lipid peroxidation?

2. line 83: what is being scavenged by the lipocalins? Do the authors think it is feasible in future work to determine directly (eg by mass spec) what kind of compounds are bound under various conditions?

3. line 109: "co-regulated". Is this up- or down-regulation or either, depending on conditions?

4. line 123: do LcoA/BncA/B orthologs occur in different phyla than the Proteobacteria?

5. Given that orthologs exist in "model" bacteria, why do the authors perform these experiments in an exotic bug like Burkholderia cenocepacia?

6. line 157: does the DPPP=O fluorescence originate from the IM, OM or both and does this matter?

7. line 174: are the cells fine at higher temperatures? What is the relevance of doing the experiments at 8 degrees C given that orthologs are also present in Enterobacteria that are unlikely to be exposed to such low temperatures?

8. line 197 and methods: regarding the homology modelling, the authors should use something more sophisticated for this, i.e. AlphaFold.

line 232: "Fig. 3B" is wrong.

9. One thing that would be useful is to check the expression of the various proteins, with and without plasmid complementation. A failure to complement could be the result of a lack of expression. Or a complementation could be the result of a 10-fold higher expression than in wild type, for example.

10. line 313: why was 0.25 MIC used and not 0.5 like in other experiments (line 280)?

11. An important issue is the selection of cells for the various fluorescence data from 5-8. How do you avoid (conscious or unconscious) cherry picking of bacteria that show what you want? The methods were not clear on this. It is really crucial to avoid selection bias since the differences between the various strains are not that big. Also, subtle differences between various panels are sometimes hard to see in the version I downloaded.

12. line 388: I assume the "antioxidant scavenging" refers to that in solution?

13. line 425: why would the loss of BcnA, LcoA and PsrA result in the accumulation of anionic phospholipids? To replace peroxidised ones?

14. line 435: "higher affinity". Higher than what?

15. Line 461: "quantitative lipidomics to determine the overall lipid content". Why don't you try and detect peroxidised lipids directly by MS? What kind of compounds do you expect? It would be great to measure peroxidation directly rather than indirectly with various fluorogenic probes.

16. line 471: replace :"under no" by "without".

17. The discussion seems overly long and could probably be condensed.

---

## [Editor Report · Decision Letter 2]

18 Mar 2022

Dear Dr Valvano,

On behalf of my colleagues and the Academic Editor, Bert van den Berg, I am pleased to say that we can in principle accept your Discovery Report "An evolutionary conserved detoxification system for membrane lipid-derived peroxyl radicals in Gram-negative bacteria" for publication in PLOS Biology, provided you address any remaining formatting and reporting issues. These will be detailed in an email that will follow this letter and that you will usually receive within 2-3 business days, during which time no action is required from you. Please note that we will not be able to formally accept your manuscript and schedule it for publication until you have completed any requested changes.

IMPORTANT: I have asked my colleagues to request that you supply the values underlying all main and supplementary figures, and to cite the location of the data clearly in each relevant main and supplementary Fig legend. 

PRESS

Sincerely, 

Dario

Dario Ummarino, PhD 

Senior Editor 

PLOS Biology

dummarino@plos.org